# Functional ANOVA for Carbon Flux Estimates from Remote Sensing Data

Jonathan Hobbs[1], Matthias Katzfuss[2], Hai Nguyen[1], Vineet Yadav[1], and Junjie Liu[1,3]

[1]Jet Propulsion Laboratory, California Institute of Technology, Pasadena, CA, USA
[2]Department of Statistics, University of Wisconsin, Madison, WI, USA
[3]California Institute of Technology, Pasadena, CA, USA

**Correspondence:** Jonathan Hobbs (Jonathan.M.Hobbs@jpl.nasa.gov)

**Abstract.** The constellation of Earth-observing satellites has now produced atmospheric greenhouse gas concentration estimates covering a period of several years. Their global coverage is providing additional information on the global carbon cycle. These products can be combined with complex inversion systems to infer the magnitude of carbon sources and sinks around the globe. Multiple factors, including the atmospheric transport model and satellite product aggregation method, can impact such flux estimates. Analysis of variance (ANOVA) is a well-established statistical framework for estimating common signals while partitioning variability across factors in the analysis of experiments. Functional ANOVA extends this approach with a statistical model that incorporates spatio-temporal correlation for each ANOVA component. The approach is illustrated on inversion experiments with different satellite retrieval aggregation methods and identifies consistent significant patterns in flux increments that span large spatial scales. Functional ANOVA identifies these patterns while accounting for the uncertainty at small spatial scales that is attributed to differences in the aggregation method. Functional ANOVA is also applied to a recent flux model intercomparison project (MIP), and the relative magnitudes of inversion system effects and data source (satellite versus in situ) are similar but exhibit slightly different importance for fluxes over different continents. In all examples, the unexplained residual variability is locally sizable in magnitude but with limited spatial and temporal correlation. These common behaviors across flux inversion experiments demonstrate the diagnostic capability for functional ANOVA to simultaneously distinguish the spatio-temporal coherence of carbon cycle processes and algorithmic factors.

## 1 Introduction

Many of the key processes in the global carbon cycle have undergone substantial change in recent decades, yet their impacts remain challenging to estimate. This is due in large part to the sparsity of direct observations of carbon fluxes. In particular, a lack of global coverage requires alternative approaches for understanding the global carbon cycle. Fluxes can be inferred indirectly with atmospheric transport models in combination with information on atmospheric carbon dioxide concentration. Regular global $CO_2$ estimates from satellites Greenhouse Gases Observing Satellite (GOSAT; Kuze et al., 2009) and the Or-

biting Carbon Observatory-2 (OCO-2; Eldering et al., 2017) have presented new challenges and opportunities for carbon cycle science investigations through flux inversions. The data volume (tens of millions of observations per year) and relatively fine spatial footprints (1.3 km × 2.25 km) for OCO-2 have motivated the use of spatially-aggregated products in global inversions (Baker et al., 2022). Since the satellite estimates arise from a retrieval (O'Dell et al., 2018), the end-to-end inference from satellite radiance spectra to flux estimates involves two complex inverse problems subject to multiple sources of uncertainty (Cressie, 2018), including observational errors, spatio-temporal representation uncertainty, and model transport error (Engelen et al., 2002). Some flux solutions attempt to account for these sources in their representation of the posterior uncertainty, but these are not always available and a coherent probabilistic assessment becomes challenging in the presence of multiple flux estimates with varying assumptions. Further, the spatio-temporal structure of flux estimates is of particular interest, and characterizing spatio-temporal correlation is necessary in quantifying uncertainty. The statistical methodology in this work provides a framework for this common situation.

Some of these sources of uncertainty, such as the data source or inversion system, can be represented as discrete instances of multiple categorical factors, and partitioning their relative contributions to the range of solutions can guide priorities for future research in carbon cycle science. In this work we are particularly interested in flux estimates derived from different inversion systems, such as those investigated in model intercomparison projects (MIPs; Thompson et al., 2016; Gaubert et al., 2019; Crowell et al., 2019). A second factor of interest is the makeup of the $CO_2$ concentration data used in the inversions. Satellite missions produce retrievals, or estimates, at the native satellite footprint resolution. Global flux inversion systems typically do not assimilate these Level 2 satellite products at this resolution. Instead, Level 3 aggregated estimates are used. Our effort contrasts inversions that assimilate a traditional simple aggregated product versus inversions that use Level 3 products produced through data fusion (Nguyen et al., 2017), which accounts for spatial correlation in the aggregation step. In the context of remote sensing data products, we define data fusion as a procedure that accounts for the spatio-temporal correlation in one or more satellite datasets to produce an estimate of a common quantity of interest at regular spatio-temporal resolution.

Given a set of flux maps obtained under different scenarios, or combinations of these factors of interest, our goal is to find common features among the scenarios, and to identify systematic ways or regions in which fluxes from different scenarios differ. Analysis of variance (ANOVA) is a statistical modeling framework that facilitates the estimation of the common and factor-specific effects. It further characterizes the magnitude of the differences within factors relative to the inherent variability within a scenario. Statistical model assumptions dictate the estimation of this within-scenario variability and will be an additional focus of our investigation. The ANOVA methodology has been extended to functional data, such as time series and spatial fields, where it can provide a coherent depiction of space/time patterns and anomalies due to various factors (Kaufman and Sain, 2010). In this functional ANOVA setting, some or all components of the classical ANOVA model are functions of space and/or time. Representing correlation of the ANOVA components across space and time is a critical extension in the functional case. While the approach and statistical model are suitable for possibly irregularly-spaced spatio-temporal data, previous implementations (Kaufman and Sain, 2010; Sain et al., 2011) have used regularly-gridded output from Earth system models, and flux inversion results are structured similarly.

The ANOVA approach can be particularly useful for analyzing output from a collection of Earth system models or assimilation systems with a common quantity of interest and similar experimental setup. This setup is often formalized as a MIP, an enterprise becoming commonplace among multi-component process models in Earth system modeling (see Eyring et al., 2016, and associated special issue). Several MIPs have been conducted for carbon flux inversion systems, both for in situ observations (Gaubert et al., 2019) and for the growing satellite record (Crowell et al., 2019; Peiro et al., 2022). These recent flux MIPs report on experiments involving multiple inversions from several modeling groups using different combinations of in situ and spatially-aggregated OCO-2 products in multiple observing modes. In addition to diagnosing differences among combinations of data sources and inversion systems, these efforts seek consensus flux estimates that suitably combine the results. Cressie et al. (2022) discuss an ANOVA-based approach with an associated statistical model to develop weights for individual results in estimating a consensus flux, and the method is demonstrated for regionally-aggregated fluxes. The current study extends this partitioning of flux variability to gridded fluxes at finer spatial resolution. This extension uses functional ANOVA and can identify the extent of spatial coherence for the factors assessed. The approach has the potential to add specificity to previous investigations into the capability of satellite data to constrain fluxes at various spatial scales (Byrne et al., 2019; Miller and Michalak, 2020).

Functional ANOVA extends the classical approach to settings with quantities of interest that are functions of known inputs such as space and/or time. The statistical model is typically extended with a specification for the relationships among the ANOVA components across space/time. For applications involving spatial fields, individual ANOVA effects are typically assumed to be spatially correlated, and this structure can be estimated from the data available. This strategy has been applied to output from regional climate models (RCMs; Kaufman and Sain, 2010; Sain et al., 2011; Kang and Cressie, 2013). Kaufman and Sain (2010) capture spatial dependence in ANOVA effects with Gaussian process (GP) models. Estimation and inference for GP models can be computationally demanding due to operations such as matrix inversion and Cholesky factorization. Sain et al. (2011) develop a Markov random field (MRF) model with a sparse precision matrix. However, in two spatial dimensions, the cost for the necessary Cholesky decomposition is $\mathcal{O}(N^{3/2})$, where $N$ is the number of locations. Another option, used in the current work, is to model the Cholesky factor of the precision matrix as sparse with the Vecchia approximation (Vecchia, 1988; Katzfuss and Guinness, 2021; Schäfer et al., 2021). This capability likely depends on geophysical and observing conditions, and the current study examines multiple regions of interest.

In this paper, we implement the functional ANOVA methodology for multiple flux inversion solutions in order to identify meaningful, spatially-coherent, carbon cycle signals and partition variability among various solutions in the multi-model ensemble. We illustrate the approach for a recent MIP effort (Peiro et al., 2022) and for flux estimates produced with multiple spatial aggregation approaches (Nguyen et al., 2020). In section 2, we describe the flux datasets used in demonstrating the functional ANOVA. Then, in section 3, we formulate the statistical model for functional ANOVA. Results are presented in section 4, and concluding remarks are provided in section 5.

## 2 Datasets

In subsequent sections, we employ the functional ANOVA methodology for multiple collections of flux inversions using in situ data and products from OCO-2. For the satellite data, the inversion systems use retrievals of $XCO_2$, the column-average dry air mole fraction of $CO_2$, which is reported in units of parts-per-million (ppm). These retrievals, termed Level 2 data products, use the Atmospheric Carbon Observations from Space (ACOS) retrieval algorithm to infer atmospheric $CO_2$ from the Level 1 satellite spectra. The examples in this paper are based on the ACOS OCO-2 Version 9 products (Kiel et al., 2019; O'Dell et al., 2018). Additional diagnostic data from the retrievals, including $XCO_2$ averaging kernels, are used in the inversions to map model states to the retrieval space when the data products are assimilated. For OCO-2 in particular, the large data volume and small spatial footprint are impractical for global flux inversions, so the retrievals are often aggregated spatially to a spatial resolution on the order of 100 km. These aggregated retrievals are examples of Level 3 products that aim to provide additional utility to a broader user community by providing manageable data volume and a regular spatio-temporal structure. In addition, aggregation can provide a more precise estimate of a quantity of interest ($CO_2$ concentration) at a coarser resolution that is still meaningful for applications. As an illustration of functional ANOVA, we investigate the impact of spatial aggregation on flux estimates in the presence of additional sources of variability.

### 2.1 Fused $CO_2$ Experiment

An ongoing NASA MEaSUREs effort aims to provide inversion-ready data products that use OCO-2 and GOSAT retrievals. The effort produces spatially-aggregated and gap-filled estimates of $XCO_2$ at daily intervals that span the period of overlap for these satellite records, from 2014 to present. The gridded OCO-2 product and multi-instrument fused product are available from the Goddard Earth Sciences Data and Information Services Center (GES DISC; Nguyen et al., 2022). The spatially aggregated $XCO_2$, along with additional quantities used in flux inversion, is estimated using a local kriging approach (Nguyen et al., 2020). The methodology accounts for and exploits the short-range spatial correlation present in the Level 2 retrievals (Torres et al., 2019; Worden et al., 2017). The spatial dependence and uncertainty associated with the aggregated $XCO_2$ are estimated from the available Level 2 retrievals and vary in space and time. The spatial aggregation also reduces the data volume, making the OCO-2 record manageable for ingestion into global flux inversion systems.

Other spatial aggregation approaches have been devised for OCO-2 inversions. The resulting data products are all structured like Level 3 products and have similar spatial resolution but differ in the underlying methodology, particularly in handling spatial correlation in the Level 2 $XCO_2$. The averaging outlined in Baker et al. (2022) has been implemented for the OCO-2 flux MIP (Peiro et al., 2022) and will be discussed in the next subsection. In addition, the NASA Carbon Monitoring System Flux (CMS-Flux) four-dimensional variational (4D-Var) inversion framework has used an aggregation approach termed "super-obs" (Liu et al., 2017; Byrne et al., 2020). The model is driven by the Goddard Earth Observing System version 5 of the NASA Global Modeling Assimilation Office (GEOS-FP) meteorology and the inversion estimates fluxes at a $4° \times 5°$ resolution. The CMS-Flux team has performed an experiment with two separate inversions: a run that ingested the traditional OCO-2 super-obs and another that ingested the MEaSUREs gridded product.

The ANOVA methodology is particularly convenient for analyzing the quantitative outcomes of experiments or trials under various discrete combinations of one or more *factors* of interest. The approach formulates a statistical model that is outlined in the next section. The parameters of the model include an overall mean response and additive effects for each combination of factors. *Replication* within factor combinations allows the decomposition of variability between (mean differences) and within combinations (noise, unexplained variability). Our demonstration of the functional ANOVA for the fused $CO_2$ experiment incorporates the data aggregation method as one experimental factor, with the two levels being the super-obs approach (Control) and the MEaSUREs product (Fused). The second factor in this investigation will be an interannual effect, contrasting June-July-August (JJA) of 2016 with the same time period in 2015.

The changing carbon cycle of the middle and high latitudes plays a critical role in the global climate system. These land areas are major carbon sinks during JJA. The multiple inversions from OCO-2 reported in Peiro et al. (2022) suggest some uncertainty in the magnitude of the biospheric uptake for JJA, particularly over Eurasia, and the spatial patterns in the summer uptake over this region have seen additional recent attention (Byrne et al., 2020). Functional ANOVA provides a framework for quantifying these spatial patterns and their interannual variability in the presence of uncertainty. The variables of interest are the CMS-Flux estimates available at monthly resolution, so individual months within a season represent replicates in this example. For spatio-temporal data applications, time is often used as a pseudo-replicate (Cressie et al., 2022; Sain et al., 2011). Figure 1 shows the collection of CMS-Flux estimates that are used in the functional ANOVA demonstration. As these maps indicate, our analysis is focused on the gridded fluxes over the land areas of Europe and Asia, specifically locations falling within the Europe, Boreal Asia, and Temperate Asia regions as defined by The Atmospheric Tracer Transport Model Intercomparison (TransCom) Project (Baker et al., 2006). The TransCom regions are used for analysis and comparisons of aggregated fluxes (Crowell et al., 2019; Cressie et al., 2022). The sign of the flux is relative to the atmosphere, so negative values indicate carbon sinks, or uptake by the land/ocean. Positive flux values represent net sources from the surface.

## 2.2 OCO-2 Flux MIP

Our second demonstration of the functional ANOVA approach comes from a multi-institution flux MIP using data products from OCO-2, which provides global estimates of $XCO_2$ suitable for assimilation into inversion systems that estimate carbon fluxes at regional to global scales. The maturity and diversity of inversion systems continue to grow with scientific interest in the carbon cycle, and a sizable collection of research groups have the capability to provide global flux estimates based on satellite data products. Like many inferences for the Earth system, flux estimation is an ill-posed inverse problem. An inversion system combines available atmospheric $CO_2$ concentration data with a transport model, prior assumptions on fluxes, and a statistical/computational inverse method. Various combinations of these system components are employed in satellite $CO_2$ inversion frameworks.

Multiple flux inversion teams applied a common inversion protocol to their individual inversion systems as part of the OCO-2 Version 9 Model Intercomparison Project (V9 MIP; Peiro et al., 2022). The MIP was designed in part to quantify the impacts of the above inversion system elements on flux estimates. In addition, each team conducted multiple inversion experiments using the same collections of atmospheric $CO_2$ data. The data collections represent combinations of in situ (IS) surface-based $CO_2$

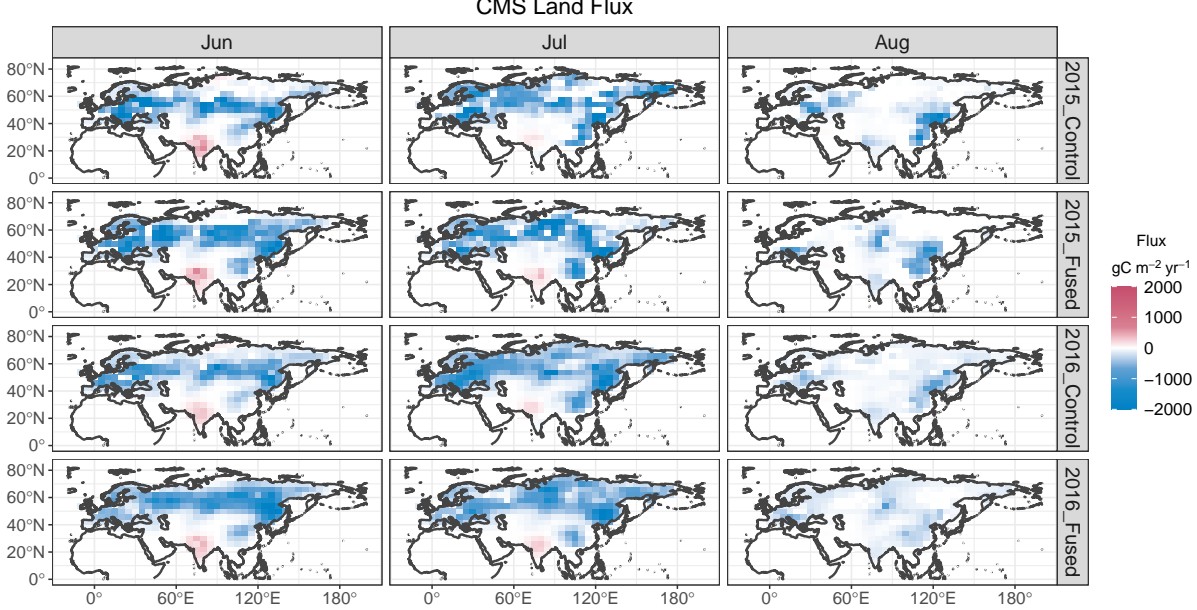

**Figure 1.** Monthly flux estimates from CMS-Flux for combinations of year and aggregation method. The top two rows depict fluxes for JJA of 2015 from the two aggregation methods, and the bottom two rows depict fluxes for JJA 2016. The Control case uses the super-obs aggregation approach and the Fused case uses the local kriging aggregation approach (Nguyen et al., 2020). Fluxes are in units of gC m$^{-2}$ yr$^{-1}$.

observations and aggregated OCO-2 retrievals from the ACOS Version 9 products (Kiel et al., 2019). The OCO-2 collections use combinations of its primary observing modes and surface types: land nadir (LN), land glint (LG), and ocean glint (OG). As noted previously, the individual retrievals are both uncertain and the associated errors are moderately correlated in space and time (Worden et al., 2017). The V9 MIP used spatially-aggregated (approximately 75 km along-track) OCO-2 retrievals following the methodology outlined in Baker et al. (2022) that partially accounts for the short-range correlation identified for the OCO-2 retrievals (Torres et al., 2019). The aggregated retrievals include uncertainty estimates that incorporate assumed spatial correlation in retrieval errors and transport uncertainty.

The V9 MIP flux experiment suite includes estimates from ten inversion systems (Peiro et al., 2022, Tables 1-2) and four combinations of data collections. Our investigation focuses on flux estimates using IS and LNLG data collections, which were also the focus in the MIP. Further, we illustrate the functional ANOVA for a subset of four inversion systems as outlined in Table 1. This subset was selected to provide a contrast among different atmospheric transport models that have similar spatial

**Table 1.** Flux inversion systems in this study; see Tables 1-2 of Peiro et al. (2022) for further details.

| Model name | Institution | Transport model | Inverse method | Reference |
|---|---|---|---|---|
| Ames | NASA Ames Research Center | GEOS-Chem | 4D-Var | Philip et al. (2019) |
| Baker | Colorado State University | PCTM | 4D-Var | Baker et al. (2010) |
| CMS-Flux | NASA Jet Propulsion Laboratory | GEOS-Chem | 4D-Var | Liu et al. (2014) |
| OU | University of Oklahoma | TM5 | 4D-Var | Crowell et al. (2018) |

resolution for the flux solution. The functional ANOVA is implemented for the spatially-referenced monthly flux estimates for JJA 2016 over North America and separately for the same time period over Africa. Figures 2 and 3 provide maps of these collections of flux estimates. In the subsequent implementation of functional ANOVA, the first factor is the flux model and the second factor is the data source (IS and LNLG). The inversion systems represented in the MIP have different approaches for estimating posterior uncertainty, if at all, and these uncertainties are not provided in the available output.

As Peiro et al. (2022) note, the global carbon cycle saw a substantial perturbation due to the El Ninño event of 2015-16, and results from a previous version of the OCO-2 MIP are also available for this period (Crowell et al., 2019). In addition to different anticipated carbon cycle responses over the two continents, North America and Africa differ substantially in coverage of in situ $CO_2$ observations. The spatial coverage of good quality OCO-2 retrievals is also limited over tropical Africa for this time period. The North America functional ANOVA implementation combines the gridded fluxes from the Boreal and Temperate North America TransCom regions. The implementation for Africa combines the Northern Africa and Southern Africa TransCom regions; the OCO-2 MIP separated Africa into a total of four sub-regions. The gridded fluxes provided by the MIP contributors are available at $1° \times 1°$ resolution.

## 3 Functional ANOVA

Analysis of variance (ANOVA) is a statistical method with a long history connected to designed experiments. In such experiments, one or more factors can be controlled at levels selected by the the experimenters, and ANOVA provides a framework for estimating the factors' impact on response variables of interest. The method relies on replication within combinations of factors, or treatments, in order to estimate a mean response for each combination of factors, along with a partitioning of variability between and within treatments. The treatment means are typically re-parameterized into an overall mean and individual effects for each level of the factors, as well as interaction effects.

The classic implementation of ANOVA considers a univariate response, such as an integrated or average carbon flux over a region of interest. This is frequently extended to a multivariate response with MANOVA, and the decomposition of variance is accompanied by estimation of the correlation structure among the multivariate responses (Johnson and Wichern, 2002). As the dimension of the multivariate response grows, the number of parameters to be estimated from the available data grows as well. The ANOVA approach can be extended to spatial fields using tools from functional data analysis and spatial statistical modeling. In this setting, the dimension can be large but the parameter space can be managed through a hierarchical approach and

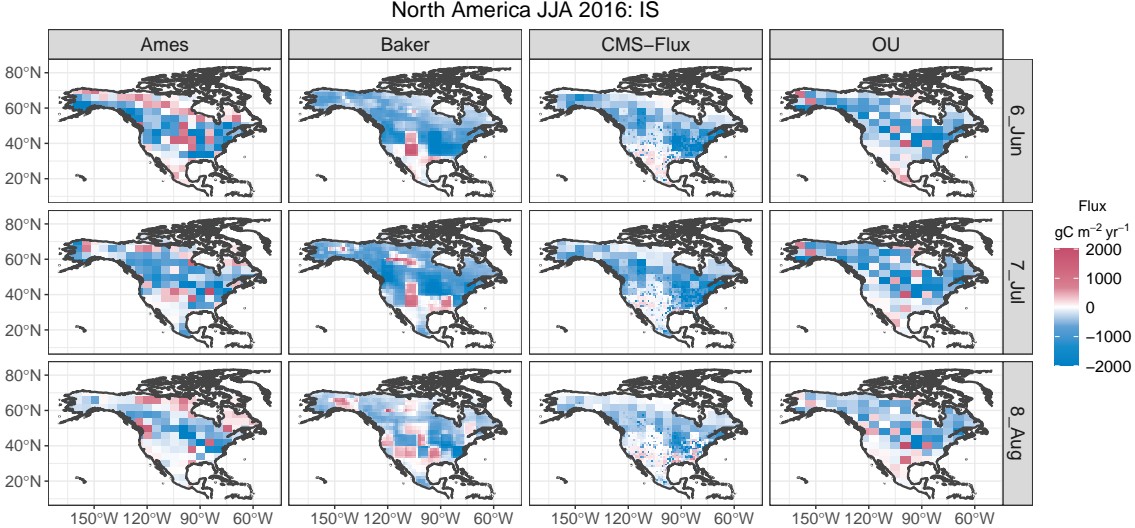

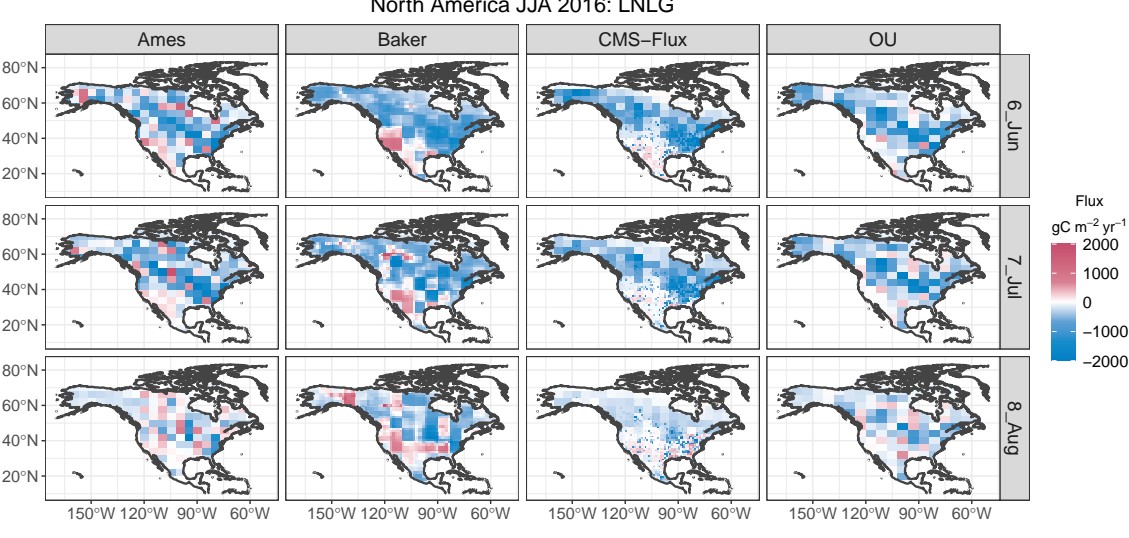

**Figure 2.** Monthly flux estimates from four inversion systems for June-July-August (JJA) 2016 over North America, with columns for the different inversions. The top three rows show flux estimates using in situ (IS) data, and the bottom three rows show flux estimates from OCO-2 land nadir and glint (LNLG) retrievals. Fluxes are in units of gC m$^{-2}$ yr$^{-1}$.

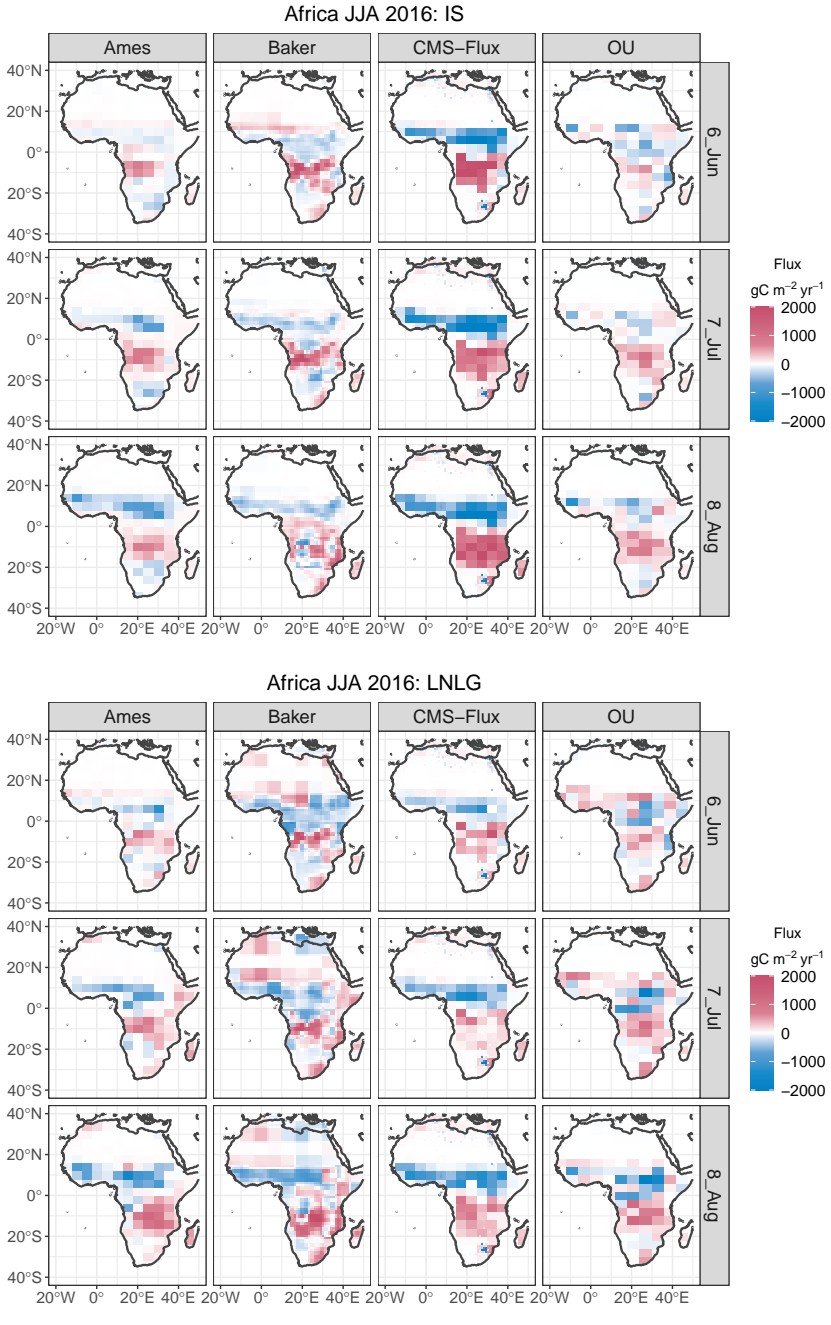

**Figure 3.** Monthly flux estimates from four inversion systems for JJA 2016 over Africa. The top three rows show flux estimates using in situ (IS) data, and the bottom three rows show flux estimates from OCO-2 land nadir and glint (LNLG) retrievals. Fluxes are in units of gC m$^{-2}$ yr$^{-1}$.

by exploiting the spatial dependence present in the data. This functional ANOVA approach has been implemented for spatial fields of output from climate model experiments (Kaufman and Sain, 2010; Kang and Cressie, 2013). Our implementation and notation for the carbon flux inversion results generally follows that from Kaufman and Sain (2010).

## 3.1 Statistical Model

In the current work, we invoke a two-way functional ANOVA in the context of carbon flux fields over land. In the two-way model, there are two experimental factors examined, generically termed Factor A and Factor B. In the fused $CO_2$ example, year is factor A, and aggregation method is factor B. In the OCO-2 flux MIP example, the modeling group is factor A, and data source is factor B. Then $y_{ijk}(\mathbf{s})$ represents the flux field at location $\mathbf{s}$ for level (setting) $i$ of factor A, level $j$ of factor B, and replicate $k$. In addition, flux inversions incorporate a space-time varying prior flux field, which we denote $y_{ijk}^{(0)}(\mathbf{s})$. The prior fluxes incorporate biospheric contributions, fossil fuel emissions, and fires. The CMS-Flux prior methodology is summarized in Liu et al. (2017) and Byrne et al. (2020), and the OCO-2 MIP prior specifications are summarized in Table 1 of Peiro et al. (2022). The functional ANOVA statistical model can be written in two equivalent forms,

$$y_{ijk}(\mathbf{s}) - y_{ijk}^{(0)}(\mathbf{s}) = \mu_{ij}(\mathbf{s}) + \epsilon_{ijk}(\mathbf{s}) \tag{1}$$

$$= \mu(\mathbf{s}) + \alpha_i(\mathbf{s}) + \beta_j(\mathbf{s}) + (\alpha\beta)_{ij}(\mathbf{s}) + \epsilon_{ijk}(\mathbf{s}). \tag{2}$$

The *cell means* formulation in (1) clearly defines a unique mean field for each treatment, $\mu_{ij}$, at level $i$ of factor A and level $j$ of factor B. With $n_\alpha$ levels of factor A and $n_\beta$ levels of factor B, there are $n_\alpha \times n_\beta$ cell mean fields. The *effects* model in (2) separates the mean field into the additive effects of the experimental factors. In the effects model, $\mu$ is the mean field representing spatial features in the common response, $\alpha_i$ quantifies the variation around $\mu$ due to level $i$ of factor A, $\beta_j$ quantifies the variation around $\mu$ due to level $j$ of factor B, and $(\alpha\beta)_{ij}$ is an interaction effect. In both forms, $\epsilon_{ijk}$ quantifies the internal variability within each scenario. In all examples, the replication within each treatment, indexed by $k$, is across months within a season. Table 2 defines the factors for the carbon flux functional ANOVA examples.

The response for the functional ANOVA models in (1) and (2) is the deviation, or flux increment, $y_{ijk}(\mathbf{s}) - y_{ijk}^{(0)}(\mathbf{s})$. This approach is employed for a combination of methodological and practical reasons. Statistical modeling for environmental applications often incorporates known information as fixed effects, leaving remaining variation to be modeled with spatio-temporal covariance structures (Cressie and Wikle, 2011). In addition, exploratory analysis of the OCO-2 V9 MIP flux estimates revealed fine-scale features in some, but not all, fluxes that were largely absent in the analyzed flux increments. The various modeling groups contributing to the MIP used different flux priors and different approaches for reporting results on the common output grid. The use of the flux increments balances a tradeoff between these complications, and this statistical modeling choice impacts the interpretation of the results of the functional ANOVA.

The effects model (2) is commonly used in ANOVA because it provides a convenient setup for partitioning variability among the factors and their interaction. Further, inference for the overall mean effect $\mu$ and linear contrasts among the factors is straightforward. However, the model is over-parameterized, so constraints among the effects must be enforced to ensure identifiability. Following Kaufman and Sain (2010), the flux inversion examples invoke sum-to-zero constraints, meaning that

**Table 2.** Summary of flux inversion results used in functional ANOVA examples. Region numbers indicate TransCom regions used in each example (e.g. Crowell et al., 2019).

| Experiment | Domain | Time Period | Factor A | Factor B |
|---|---|---|---|---|
| Records of Fused $CO_2$ | Eurasia | JJA 2015, 2016 | Year | Aggregation Method |
| | Regions 7, 8, 11 | | $n_\alpha = 2$ | $n_\beta = 2$ |
| OCO-2 V9 MIP | North America | JJA 2016 | Inversion System | Data Source |
| | Regions 1, 2 | | $n_\alpha = 4$ | $n_\beta = 2$ |
| OCO-2 V9 MIP | Africa | JJA 2016 | Inversion System | Data Source |
| | Regions 5, 6 | | $n_\alpha = 4$ | $n_\beta = 2$ |

the factor main and interaction effects add to zero, i.e.

$$\sum_{i=1}^{n_\alpha} \alpha_i(\mathbf{s}) = 0, \qquad \sum_{j=1}^{n_\beta} \beta_j(\mathbf{s}) = 0,$$

for all locations $\mathbf{s}$. For interaction effects, the constraints apply across all levels of each factor,

$$\sum_{i=1}^{n_\alpha} (\alpha\beta)_{ij}(\mathbf{s}) = 0; \quad j = 1, \ldots, n_\beta, \qquad \sum_{j=1}^{n_\beta} (\alpha\beta)_{ij}(\mathbf{s}) = 0; \quad i = 1, \ldots, n_\alpha.$$

In classic univariate ANOVA, the effects model parameters are estimated by assembling a series of contrast effects of reduced dimension to ensure identifiability. For factor A, there are $i' = 1, \ldots, n_\alpha - 1$ of these contrast effects, denoted $\alpha_{i'}^*$. In the case where $n_\alpha = 2$, there is a single contrast effect, which can be interpreted as the difference in mean response for the two levels of Factor A; in the fused $CO_2$ example, this contrast effect is the difference between the two years studied, 2016 and 2015. Similarly, factor B has $j' = 1, \ldots, n_\beta - 1$ contrast effects $\beta_{j'}^*$, and there are $(n_\alpha - 1) \times (n_\beta - 1)$ contrast effects for interaction,
$(\alpha\beta)_{i'j'}^*$. The contrast effects are related to the original effects model (2) parameters through linear transformations. This is discussed further in the supplement (Section S.1.2) and in Kaufman and Sain (2010).

Since the quantities of interest are spatial fields, the ANOVA effects are functions of location. The estimation can account for this structure and exploit potential spatial correlation if a suitable spatial statistical model is incorporated in a hierarchical fashion. To that end, a Gaussian process (GP) is assumed for each spatial field. In this study, the flux inversion results are
reported on a spatial grid. For each ANOVA component, the collection of $N$ locations has a joint multivariate Gaussian distribution. The vector $\boldsymbol{\mu} = \{\mu(\mathbf{s}_\ell) : \ell = 1, \ldots N\}$ includes the ANOVA overall mean for the observed locations. Then

$$\boldsymbol{\mu} \sim \mathcal{N}(\mathbf{0}, \boldsymbol{\Sigma}_\mu).$$

The covariance matrix $\boldsymbol{\Sigma}_\mu$ captures the spatial dependence for the ANOVA overall mean components across locations. For the carbon flux inversion examples, we use the Matérn covariance model with parameters $\boldsymbol{\theta}_\mu \equiv \{\sigma_\mu, \lambda_\mu, \nu_\mu\}$, where $\sigma_\mu$ is
a standard deviation, $\lambda_\mu$ is a range parameter describing the rate of decay of spatial correlation with distance, and $\nu_\mu$ is a smoothness parameter (Stein, 1999). For an exponential model with smoothness $\nu = 0.5$, the range parameter is the distance

at which the correlation reaches $1/e$. When the smoothness parameter is larger, the correlation decays more slowly at short distances. The GP mean here is taken to be zero because the flux prior has been subtracted in (1)-(2).

Analogous GP assumptions are made for the remaining ANOVA model components. When $n_\alpha > 2$ or $n_\beta > 2$, some components have multiple realizations, which are assumed to be independent and identically distributed (iid) GP realizations. The GP assumptions are applied to the contrast effect specification of the main effects and interactions,

$$\boldsymbol{\alpha}_{i'}^* \sim \mathcal{N}(\mathbf{0}, \boldsymbol{\Sigma}_\alpha); \quad i' = 1, \ldots, n_\alpha - 1$$

$$\boldsymbol{\beta}_{j'}^* \sim \mathcal{N}(\mathbf{0}, \boldsymbol{\Sigma}_\beta); \quad j' = 1, \ldots, n_\beta - 1$$

$$(\boldsymbol{\alpha\beta})_{i'j'}^* \sim \mathcal{N}(\mathbf{0}, \boldsymbol{\Sigma}_{\alpha\beta}); \quad i' = 1, \ldots, n_\alpha - 1; \quad j' = 1, \ldots, n_\beta - 1$$

Functional ANOVA for geophysical model output has often used different time points to yield multiple pseudo-replicates $k = 1, \ldots, n_\epsilon$ across the combinations of ANOVA factors (Kaufman and Sain, 2010; Sain et al., 2011). In these investigations, these replicates represent different years in a multi-decade climate model run, and the functional ANOVA models assumed dependence across space but independence over time for $\epsilon_{ijk}$. In the current investigation, the replicates are consecutive months within a season, and independence across time may not be a reasonable assumption. Instead, potential correlation across replicates within a season is represented with a first-order autoregressive (AR1) structure. This specification is analogous to a repeated measures model in classical ANOVA (Johnson and Wichern, 2002). We define a stacked error vector for all replicates within a factor combination,

$$\boldsymbol{\epsilon}_{ij} \equiv \begin{bmatrix} \boldsymbol{\epsilon}_{ij1} \\ \cdots \\ \boldsymbol{\epsilon}_{ijn_\epsilon} \end{bmatrix}.$$

The spatio-temporal covariance for the error process is

$$\boldsymbol{\epsilon}_{ij} \sim \mathcal{N}(\mathbf{0}, \boldsymbol{\Sigma}_\epsilon); \quad i = 1, \ldots, n_\alpha; \, j = 1, \ldots, n_\beta;$$

The spatio-temporal covariance $\boldsymbol{\Sigma}_\epsilon$ follows a Kronecker product form,

$$\boldsymbol{\Sigma}_\epsilon = \mathbf{R}_{\boldsymbol{\theta}_\epsilon} \otimes \mathbf{C}_{\boldsymbol{\theta}_\epsilon},$$

where $\mathbf{C}_{\boldsymbol{\theta}_\epsilon}$ is the $N \times N$ spatial covariance matrix and $\mathbf{R}_{\boldsymbol{\theta}_\epsilon}$ is the temporal correlation matrix parameterized by the AR correlation $\rho_\epsilon$. For the case of $n_\epsilon = 3$ consecutive months in the flux inversion examples, the resulting temporal correlation matrix has the form

$$\mathbf{R}_{\boldsymbol{\theta}_\epsilon} = \begin{bmatrix} 1 & \rho_\epsilon & \rho_\epsilon^2 \\ \rho_\epsilon & 1 & \rho_\epsilon \\ \rho_\epsilon^2 & \rho_\epsilon & 1 \end{bmatrix}$$

The full collection of covariance parameters for the error process is $\boldsymbol{\theta}_\epsilon \equiv \{\sigma_\epsilon, \lambda_\epsilon, \nu_\epsilon, \rho_\epsilon\}$.

## 3.2 Estimation

Bayesian inference is commonly used for hierarchical spatio-temporal statistical models and has been implemented in previous work on functional ANOVA incorporating spatial dependence (Kaufman and Sain, 2010; Kang and Cressie, 2013). The computational overhead for Bayesian inference can be substantial, particularly when working with GP models for spatial processes, since operations must be performed on large, dense covariance matrices numerous times. Therefore, we formulate the GP models in the functional ANOVA using the Vecchia approximation of Katzfuss and Guinness (2021) and Schäfer et al. (2021). The representation yields sparse matrices that allow for more efficient computations in Bayesian inference. The GP approximations are still functions of the same parameters as the original Matérn models, and each ANOVA component has a unique set of parameters. Bayesian inference interrogates the joint posterior distribution of the ANOVA components and parameters, $p()$, given the available flux fields. This is written as

$$p\left(\boldsymbol{\mu}, \boldsymbol{\alpha}^*, \boldsymbol{\beta}^*, (\boldsymbol{\alpha\beta})^*, \boldsymbol{\theta} | \boldsymbol{y} - \boldsymbol{y}^{(0)}\right) \propto f\left(\boldsymbol{y} - \boldsymbol{y}^{(0)} | \boldsymbol{\mu}, \boldsymbol{\alpha}^*, \boldsymbol{\beta}^*, (\boldsymbol{\alpha\beta})^*, \boldsymbol{\theta}\right) f\left(\boldsymbol{\mu}, \boldsymbol{\alpha}^*, \boldsymbol{\beta}^*, (\boldsymbol{\alpha\beta})^* | \boldsymbol{\theta}\right) \pi(\boldsymbol{\theta}), \tag{3}$$

where $f(\boldsymbol{y} - \boldsymbol{y}^{(0)} | \cdot)$ is a joint Gaussian likelihood arising from the GP models for the ANOVA components, including the noise, and $\pi(\boldsymbol{\theta})$ is a prior distribution for the collection of GP parameters, $\boldsymbol{\theta} \equiv \{\boldsymbol{\theta}_\mu, \boldsymbol{\theta}_\alpha, \boldsymbol{\theta}_\beta, \boldsymbol{\theta}_{(\alpha\beta)}, \boldsymbol{\theta}_\epsilon\}$. Prior distributions are independent for all elements of $\boldsymbol{\theta}$. The distributional forms and parameters are selected to maintain proper, yet high-variance, prior distributions guided by previous work on Bayesian analysis of hierarchical models (Gelman, 2006; Kaufman and Sain, 2010). Further details on the prior distribution assumptions are provided in the supplement (Figure S.1).

The posterior distribution (3) is complex and high-dimensional, but it can be sampled using Markov chain Monte Carlo (MCMC) methods. In particular, a Metropolis-within-Gibbs MCMC algorithm is used (Gelman et al., 2013). This approach uses the general Gibbs sampler to sample sequentially at each iteration from individual-component conditional posterior distributions, $p(\boldsymbol{\mu} | \boldsymbol{\theta}_\mu, \boldsymbol{\theta}_\epsilon, \boldsymbol{y} - \boldsymbol{y}^{(0)})$, $p(\boldsymbol{\alpha}^* | \boldsymbol{\theta}_\alpha, \boldsymbol{\theta}_\epsilon, \boldsymbol{y} - \boldsymbol{y}^{(0)})$, $p(\boldsymbol{\beta}^* | \boldsymbol{\theta}_\beta, \boldsymbol{\theta}_\epsilon, \boldsymbol{y} - \boldsymbol{y}^{(0)})$, and $p((\boldsymbol{\alpha\beta})^* | \boldsymbol{\theta}_{\alpha\beta}, \boldsymbol{\theta}_\epsilon, \boldsymbol{y} - \boldsymbol{y}^{(0)})$. As outlined in Section S.1.3 of the supplement, the individual component distributions are multivariate Gaussian and depend on summary statistics of the data $\boldsymbol{y} - \boldsymbol{y}^{(0)}$, the GP parameters for the component being updated, and the noise, but not on other ANOVA components. In addition to these distributions, the Gibbs sampler cycles through draws from the GP parameters' conditional distributions: $p(\boldsymbol{\theta}_\mu | \boldsymbol{\mu})$, $p(\boldsymbol{\theta}_\alpha | \boldsymbol{\alpha}^*)$, $p(\boldsymbol{\theta}_\beta | \boldsymbol{\beta}^*)$, $p(\boldsymbol{\theta}_{\alpha\beta} | (\boldsymbol{\alpha\beta})^*)$, and $p(\boldsymbol{\theta}_\epsilon | (\boldsymbol{y} - \boldsymbol{y}^{(0)})$. Each distribution is sampled with a Metropolis-Hastings (MH) proposal. Further details on the MCMC procedure can be found in the supplement (Section S.1.3).

The functional ANOVA model and MCMC algorithm for the carbon flux examples are broadly similar to previous demonstrations with climate model output (Kaufman and Sain, 2010; Sain et al., 2011; Kang and Cressie, 2013), but there are a few notable extensions in the current work. The OCO-2 MIP examples use more than two levels per factor, which is addressed through the contrast effects. All examples include Bayesian inference for all Matérn parameters; in previous work the smoothness parameter $\nu$ was often fixed. The current implementation includes temporal correlation for the ANOVA error term to handle pseudo-replicates at adjacent times. Finally, the current work invokes the Vecchia approximation (Katzfuss and Guinness, 2021) for the Gaussian processes, which which makes the MCMC algorithm more computationally efficient.

## 4 Results

The MCMC algorithm outlined in the previous section yields a large collection of random draws from the posterior distribution of the spatio-temporal covariance parameters and ANOVA components. The posterior samples can be summarized for individual parameters, as well as for arbitrary functions of them. For example, while the MCMC samples the contrast effects $\alpha^*$, the draws can be transformed to summarize the main effects $\alpha$. The functional ANOVA results for the experiments outlined in Table 2 are summarized in various ways in this section. As outlined in (2), inference for the ANOVA components is carried on flux increments, so they are interpreted in a relative sense.

It is also worth emphasizing that the inference can be broadly partitioned into two categories of quantities. The first category includes the covariance parameters $\boldsymbol{\theta}$. These results inform the overall collective behavior of the ANOVA factors across levels. In addition to partitioning variability with variance components through the GP standard deviation $\sigma$ as in classical ANOVA (Johnson and Wichern, 2002), the functional ANOVA inference also characterizes the spatio-temporal coherence, particularly through the range $\lambda$ and autocorrelation $\rho$, for each factor. The magnitudes of these covariance parameters across factors is meaningful in diagnosing the relative impact of their respective sources of uncertainty in the flux inversions. The second broad category of inference includes the ANOVA components, $(\mu, \alpha^*, \beta^*, (\alpha\beta)^*)$. These inferences provide specific information about the spatial patterns of the ANOVA components.

### 4.1 Fused $CO_2$ Experiment

The CMS-Flux inversion results over Eurasia for JJA 2015 and 2016 using the super-obs and data fusion aggregation methods were incorporated into the first functional ANOVA implementation. Table 3 summarizes the posterior distributions for the GP parameters for each of the ANOVA components. The estimated GP standard deviation is largest for the error fields $\epsilon$, indicating that the month-to-month variability within each treatment combination is relatively large. However, the estimated range parameter $\lambda_\epsilon$ is relatively small. The estimated range for the overall mean $\mu$ and year effect $\alpha$ exceed 1000 km and are an order of magnitude larger. These contrasts in the estimated correlation range are meaningful in multiple respects. The large range values for the multi-year mean flux increment and the year effect indicate these factors have coherent regional-to continental-scale patterns at the seasonal timescale. These components, $\mu$ and $\alpha$, are connected to intrinsic behavior of the carbon cycle as represented from this collection of inversions. In contrast, the aggregation method effect, $\beta$, shows differences that have limited spatial coherence. Notably, the posterior inference shows no evidence of the error process AR correlation $\rho_\epsilon$ being different from zero. The error term, which captures both intraseasonal plus other unexplained variation, also has limited spatial coherence. Relatively large unexplained variability at these small scales may be due, in part, to the inherent ill-posed nature of the flux inversion.

The contrast in spatial coherence among the ANOVA components is also evident in the location-specific posterior means of the ANOVA components, which are summarized in Figure 4. The upper left panel provides the estimates of $\mu(\mathbf{s})$, the overall mean deviation from the prior flux for JJA across the two years, 2015-16. There are broad swaths of the domain with sizable negative mean effects, particularly over central and eastern Asia. The estimated difference between 2016 and 2015, $\alpha^*(\mathbf{s})$,

**Table 3.** Functional ANOVA spatio-temporal covariance parameter estimates for the CMS-Flux inversions in the fused $CO_2$ experiment. The posterior median is the value listed first in each cell, and the values in parentheses are the lower and upper endpoints of 95% credible intervals. Data and standard deviations have units of gC m$^{-2}$ yr$^{-1}$. Range parameters have units of km and smoothness parameters are unitless.

CMS-Flux Fusion Experiment, Eurasia JJA 2015-16

| Parameter | Mean $\mu$ | Year $\alpha$ 2016 - 2015 | Aggregation $\beta$ Fused - SuperObs | Interaction $(\alpha\beta)$ | Error $\epsilon$ |
|---|---|---|---|---|---|
| Standard Deviation $\sigma$ | 137 (109, 185) | 5.1 (0.5, 22.7) | 3.3 (0.1, 12.4) | 2.7 (0.2, 10.0) | 298 (289, 307) |
| Range $\lambda$ [km] | 740 (460, 1420) | 600 (80, 4800) | 160 (50, 410) | 150 (60, 410) | 295 (259, 337) |
| Smoothness $\nu$ | 1.07 (0.73, 1.57) | 1.00 (0.52, 1.71) | 0.94 (0.51, 1.57) | 0.91 (0.49, 1.54) | 1.05 (0.91, 1.21) |
| AR Correlation $\rho$ | | | | | -0.006 (-0.025, 0.013) |

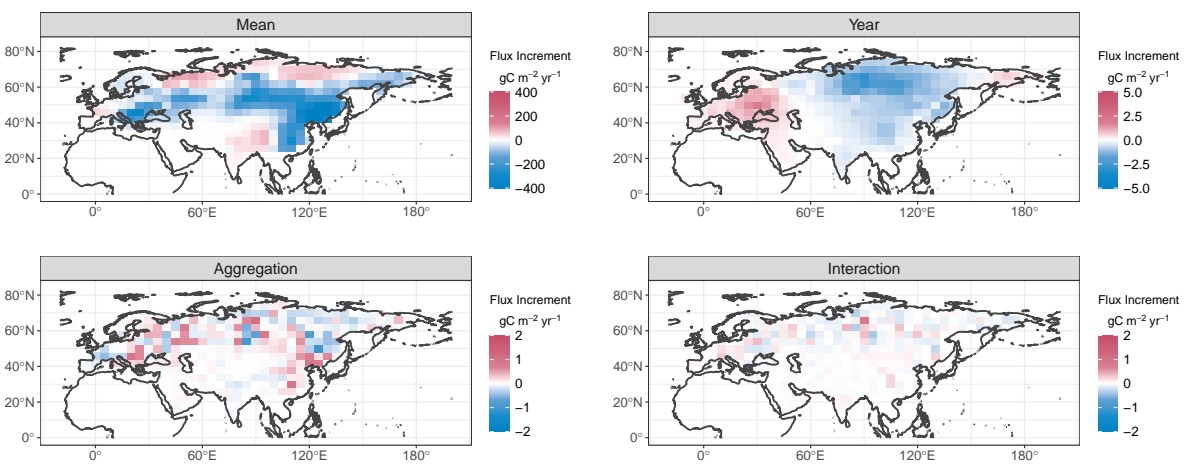

**Figure 4.** Posterior means for functional ANOVA model components for the records of fused $CO_2$ example. Note the different color scales for the panels.

shown in the upper right panel also exhibits large-scale coherence but of a modest magnitude. The contrast effects for the aggregation method, $\beta^*(\mathbf{s})$, and for interaction, $(\alpha\beta)^*(\mathbf{s})$, are shown in the bottom left and right panels, respectively. In both cases, the magnitudes are small with limited spatial coherence.

For this example, there is particular interest in the impact of the aggregation method on the estimated fluxes from an inversion system. Figure 5 illustrates a quantity from the posterior distribution for detecting meaningful signals in the flux increments while accounting for uncertainty due to different aggregation methods. The left panel shows $Pr(|\mu(\mathbf{s})| > |\beta^*(\mathbf{s})|)$, the location-specific posterior probability that the overall mean flux increment has a magnitude greater than the magnitude of the effect due to aggregation method. The MCMC posterior samples include realizations for the ANOVA components at all locations, so these probabilities can be estimated from these samples. The probabilities are nearly 1 for the entire domain, with some slight regional

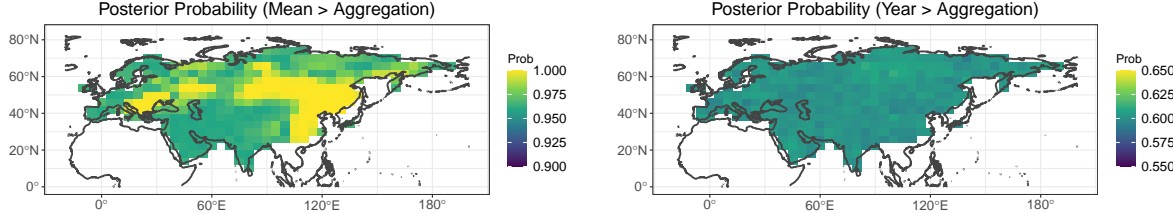

**Figure 5.** Posterior probabilities for the records of fused $CO_2$ example. Left panel shows the probability that the magnitude of the overall mean exceeds the aggregation method effect, $Pr(|\mu(\mathbf{s})| > |\beta^*(\mathbf{s})|)$. Right panel shows the probability that the magnitude of the year effect exceeds the aggregation method effect, $Pr(|\alpha^*(\mathbf{s})| > |\beta^*(\mathbf{s})|)$

differences. This suggests that the overall mean flux increments are meaningfully distinguishable from differences due to the use of the fused product versus the super-obs approach. From a practical standpoint, the impact of the aggregation approach is insignificant relative to the overall mean signal. Similarly, the right panel of Figure 5 shows $Pr(|\alpha^*(\mathbf{s})| > |\beta^*(\mathbf{s})|)$, the posterior probability that the year-to-year difference has a magnitude greater than the aggregation effect. These probabilities are fairly uniform spatially at around 0.6. Taken together, these conclusions highlight that persistent flux increments are identified in the presence of data aggregation uncertainty while more subtle interannual fluctuations are not as clearly identified.

## 4.2 OCO-2 Flux MIP

The functional ANOVA inference was carried out separately for flux fields from the OCO-2 V9 flux MIP over North America and Africa for JJA 2016. In both cases the two ANOVA factors are the flux inversion system with four levels (modeling groups) and the data source with two levels (IS and LNLG). These two regions represent distinct scenarios for the methodology for a number of reasons. The carbon cycle of the temperate and boreal land regions, and transitions therein, of North America differ from the tropical and subtropical areas of Africa. In addition, data availability for the two regions is markedly different. As shown in Figure 1 of Peiro et al. (2022), the density of in situ $CO_2$ observations is substantially higher over North America than over Africa. OCO-2 has dense coverage over both continents with some regional disparities (O'Dell et al., 2018). For example, OCO-2 has substantially more successful retrievals over northern and southern Africa than over the tropics. Despite some of these differences in data coverage, both continents should exhibit some spatial heterogeneity in the overall flux signal.

Table 4 summarizes the posterior distributions for the GP parameters for the OCO-2 MIP functional ANOVA. Once again, the estimated GP standard deviation $\sigma_\epsilon$ is largest for the error fields $\epsilon$ for both regions, indicating sizable month-to-month variability after accounting for the overall mean increment along with the model, data source, and interaction effects. In addition, the spatial range $\lambda_\epsilon$ is relatively short for the error component. The estimated spatial range is largest among all components for the flux model effect ($\lambda_\alpha$) for both regions. This result is consistent with large-scale regional flux differences among inversion systems with different driving atmospheric transport (Peiro et al., 2022; Basu et al., 2018). Further, since the statistical model is applied to the flux increments, prior flux differences contribute to the model effects.

**Table 4.** Functional ANOVA spatio-temporal covariance parameter estimates for North America and Africa for JJA 2016. The posterior median is the value listed first in each cell, and the values in parentheses are the lower and upper endpoints of 95% credible intervals. Data and standard deviations have units of gC m$^{-2}$ yr$^{-1}$. Range parameters have units of km and smoothness parameters are unitless.

OCO-2 MIP, North America JJA 2016

| Parameter | Mean $\mu$ | Flux Model $\alpha$ | Data Source $\beta$ | Interaction $(\alpha\beta)$ | Error $\epsilon$ |
|---|---|---|---|---|---|
| Standard Deviation $\sigma$ | 150 (134, 171) | 22 (15, 42) | 4.2 (0.2, 32.5) | 1.3 (0.5, 4.9) | 398 (393, 403) |
| Range $\lambda$ [km] | 202 (162, 257) | 2400 (900, 8700) | 220 (80, 710) | 160 (60, 410) | 107 (104, 111) |
| Smoothness $\nu$ | 1.11 (1.00, 1.23) | 1.07 (0.80, 1.43) | 1.23 (0.67, 1.90) | 0.97 (0.47, 1.46) | 1.26 (1.23, 1.28) |
| AR Correlation $\rho$ | | | | | 0.130 (0.126, 0.135) |

OCO-2 MIP, Africa JJA 2016

| Parameter | Mean $\mu$ | Flux Model $\alpha$ | Data Source $\beta$ | Interaction $(\alpha\beta)$ | Error $\epsilon$ |
|---|---|---|---|---|---|
| Standard Deviation $\sigma$ | 66 (58, 76) | 5.3 (0.4, 11.2) | 26 (18, 39) | 7.6 (0.2, 14.5) | 193 (191, 196) |
| Range $\lambda$ [km] | 223 (173, 294) | 820 (390, 2820) | 520 (350, 810) | 540 (160, 890) | 72.6 (69.5, 75.7) |
| Smoothness $\nu$ | 1.42 (1.22, 1.66) | 1.21 (0.41, 1.71) | 1.70 (1.28, 2.18) | 1.64 (0.81, 2.20) | 2.87 (2.74, 3.02) |
| AR Correlation $\rho$ | | | | | 0.147 (0.142, 0.152) |

The two regions differ slightly in the relative variability of the model, data source, and interaction effects $(\sigma_\alpha, \sigma_\beta, \sigma_{(\alpha\beta)})$. Over North America, the model and data source effects have similar variability, and the interaction effect is an order of magnitude smaller. On the other hand, the variability in the model and interaction effects are similar over Africa, but the data source effect standard deviation, $\sigma_\beta$, is larger than these. The difference in coverage between in situ and OCO-2 likely contributes to this relatively large data source effect. For both MIP examples, the temporal autocorrelation for the error process is positive

but relatively small in magnitude at less than 0.15.

     Noting again that the analysis is carried out on the inversions' deviations from their respective priors, Figure 6 shows the estimated spatial pattern for the overall mean flux increment, $\mu(\mathbf{s})$, for the OCO-2 MIP North America functional ANOVA. The patterns indicate generally negative increments (increased uptake) over the eastern United States and positive increments over the southwestern U.S. While some spatial coherence is present, the estimated range is around 200 km and less than that

for the model effect in particular. This difference is evident in Figure 7, which provides maps of the estimated (posterior mean) individual model effects, $\alpha(\mathbf{s})$, as well as the data source effects, $\beta(\mathbf{s})$. While generally smaller in magnitude than some of the localized mean increments, the estimated model effects in the top panels of Figure 7 are quite coherent across the continent for each model. Since this component reflects model-to-model differences in flux increments, the patterns can be a combination of difference in prior fluxes, as well as other aspects of the inversions, particularly atmospheric transport. The data source effect

exhibits spatial dependence that decays at shorter distances. The data source effect maps also include the locations of in situ observations from the *obspack_co2_1_GLOBALVIEWplus_v8.0_2022-08-27* dataset (Cooperative Global Atmospheric Data Integration Project, 2022).

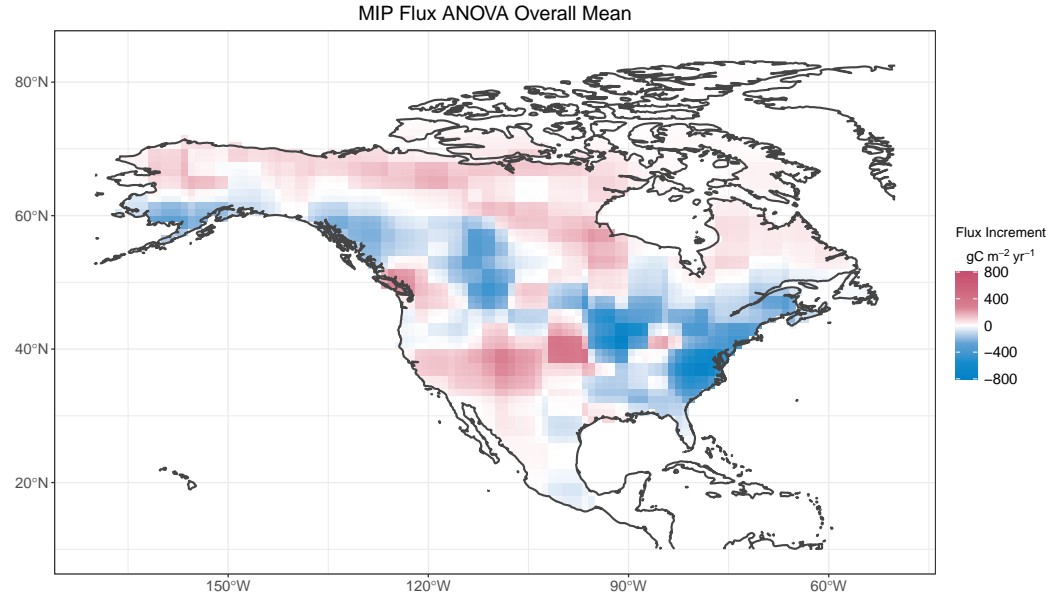

**Figure 6.** Posterior mean for the functional ANOVA overall mean flux increment $\mu$ for JJA 2016 over North America.

The MCMC procedure provides samples from the full joint posterior distribution, and the samples can be summarized in various ways to describe the uncertainty for quantities of interest. Figure 8 summarizes the posterior distribution for the overall
mean increment $\mu(\mathbf{s})$ through spatially-referenced credible intervals. Some of the inferred local increments are evident here, including negative values over the U.S. Midwest and Atlantic coast, and modest positive changes over the Southwest. Over much of the domain, the intervals do cover zero, meaning that the direction of the mean flux increment estimated by the posterior is ambiguous.

Figure 9 shows the estimated spatial pattern for the overall mean flux increment, $\mu(\mathbf{s})$, for the OCO-2 MIP Africa functional
ANOVA. This posterior mean map suggests a broad negative increment over western tropical Africa. The prior fluxes over the continent (see Figure S.3) contrast uptake north of the Equator and a net source to the south. This contrast manifests in some of the remaining ANOVA effects, as shown in Figure 10, where the estimated model effects (top panels) change sign across the Equator. It should also be noted that the magnitude of these model effects is generally smaller over Africa than over North America. A north-south contrast is also evident in the data source effect estimates in the lower panel of Figure
10, which could relate to inter-hemispheric transport differences among the inversion systems. The contrast between the two data sources (IS - LNLG) is captured in the contrast effect $\beta^*(\mathbf{s})$. Posterior credible intervals for this contrast are mapped in Figure 11. For most of the continent the intervals cover zero, but the in situ inversions appear to have consistently higher fluxes over southern tropical Africa. This area is notably devoid of in situ observation sites. At the same, the density of quality

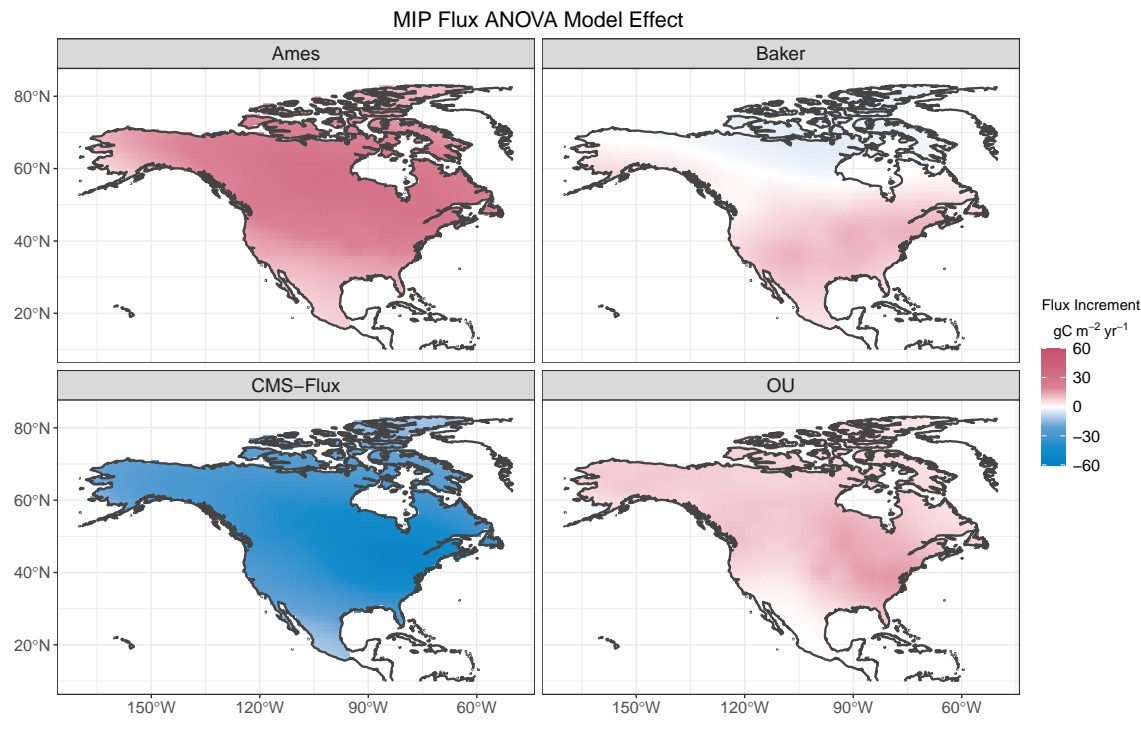

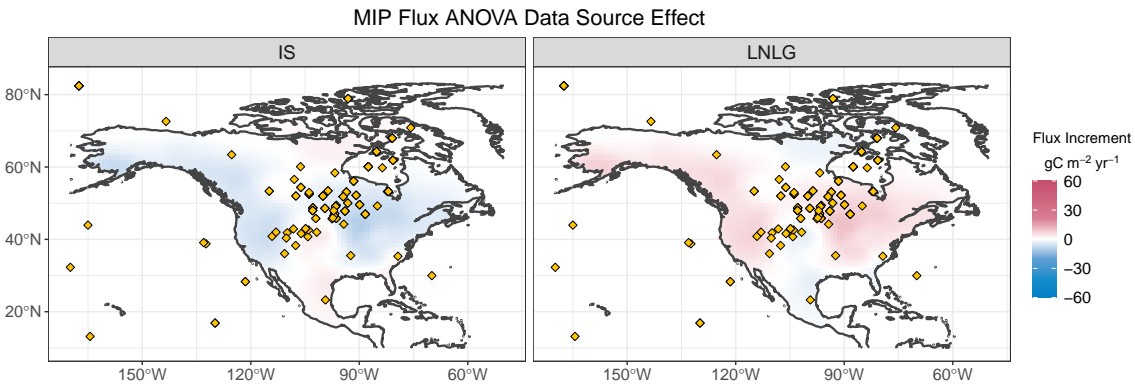

**Figure 7.** Posterior mean for the functional ANOVA main effect for flux model ($\alpha$, top) and for data source ($\beta$, bottom) for JJA 2016 over North America. In situ observation locations are indicated by gold diamonds.

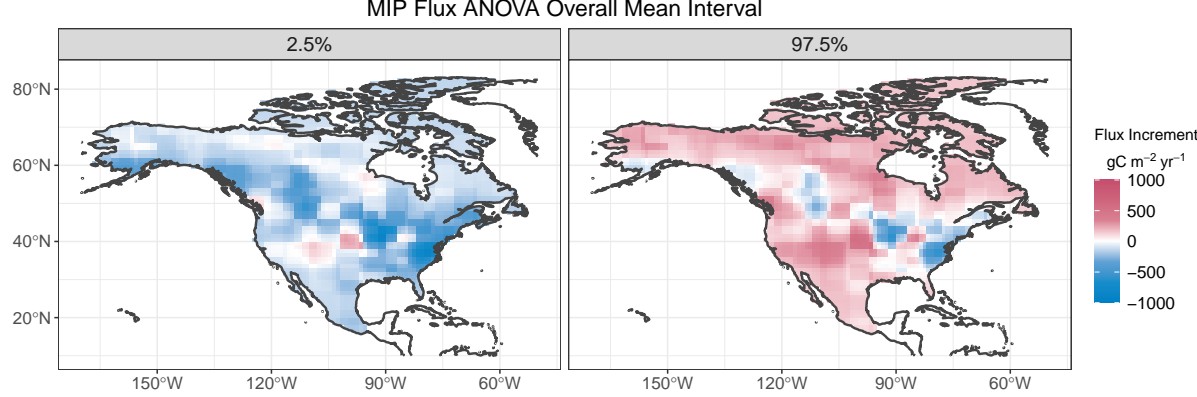

**Figure 8.** Posterior credible intervals for the functional ANOVA overall mean $\mu$ for JJA 2016 over North America. Left (right) panel depicts the 2.5th (97.5th) percentile of the posterior distribution for each location.

OCO-2 retrievals is diminished near the Equator over Africa as well, and data density is more substantial over the northern portion of the continent (see Figure A1 of Peiro et al., 2022). These discrepancies for a data-limited region underscore the role of representing atmospheric transport accurately and will be a challenge for a region that is susceptible to substantial carbon-climate perturbations (Liu et al., 2017).

## 5 Conclusions

Flux inversions produce estimates of the land-ocean-atmosphere exchange of carbon as spatio-temporal fields, providing critical information on the global climate system. These estimates can be variable due to the combinations of factors, such as the atmospheric transport model and the input data source(s), used in the inversions. Recent application of classical analysis of variance (ANOVA) to spatially-aggregated fluxes provided a statistical model for partitioning variability among these factors while estimating the common signals from the various flux inversions (Cressie et al., 2022). The ANOVA estimates provided an empirical approach for weighting different inversion systems in a MIP, as well as an efficient consensus flux estimate. The functional ANOVA presented and illustrated here extends the approach to a large collection of gridded fluxes at finer spatial resolution. The statistical model represents the flux increment as a function of space and time, with additive spatial fields for an overall mean, main effects for each factor, and their interactions. Remaining variability is captured in a spatio-temporal error component. In doing so, the approach accounts for spatial dependence in the flux fields. The extent of spatial dependence

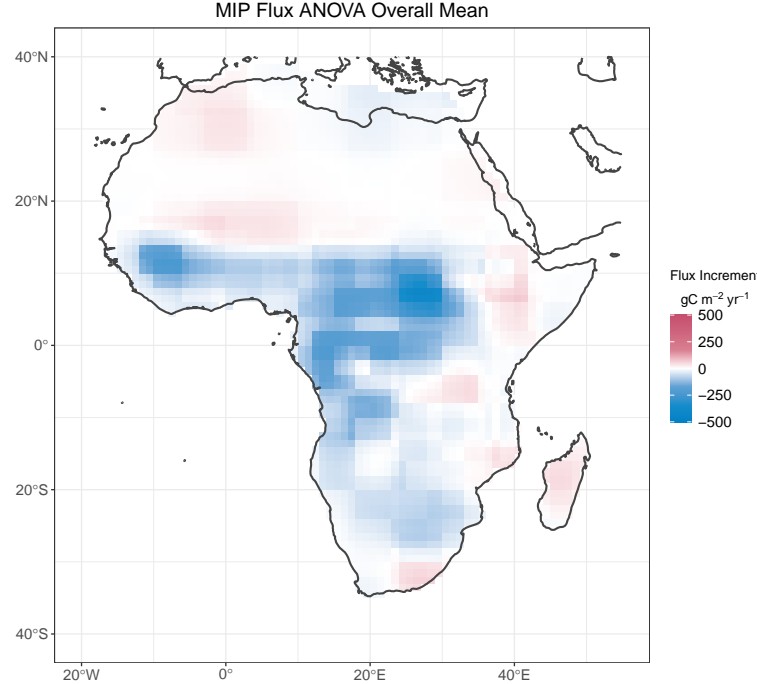

**Figure 9.** Posterior mean for the functional ANOVA overall mean flux increment $\mu$ for JJA 2016 over Africa.

is estimated separately for each of the factors considered, along with their interactions. Each of the ANOVA components is
represented as a spatial GP using the Vecchia approximation for computational efficiency. In demonstrating the functional
ANOVA, our objectives differ slightly from Cressie et al. (2022). This paper has illustrated the functional ANOVA method
for flux estimates at continental scale under multiple configurations (Table 2) and has emphasized the method's capability to
partition variability and discriminate spatial coherence across the different factors considered.

The CMS-Flux inversion system was used in a set of inversion experiments to investigate the impact of satellite retrieval
aggregation on flux inferences, contrasting a super-obs method with data fusion for aggregating fine-scale OCO-2 retrievals.
Overall the aggregation method effect estimated via functional ANOVA was small in magnitude and in its extent of spatial
dependence, particularly relative to the overall mean flux increment. Estimated flux differences across years exhibit substantial
spatial coherence relative to the aggregation method and interaction effects. At the same time, the interannual variability is of
a similar order of magnitude as the aggregation method effect, so detecting year-to-year differences is more challenging with a
limited number of pseudo-replicates.

The functional ANOVA was also implemented for a subset of inversions from the OCO-2 flux MIP over both North America
and Africa for JJA 2016. The functional ANOVA identified local consensus in flux increments for both continents in the
presence of variability across inversion systems and atmospheric $CO_2$ data sources. Over Africa, the data source effect (in situ
versus satellite-only) was larger than the flux model effect. This assessment could be a useful diagnostic for understanding

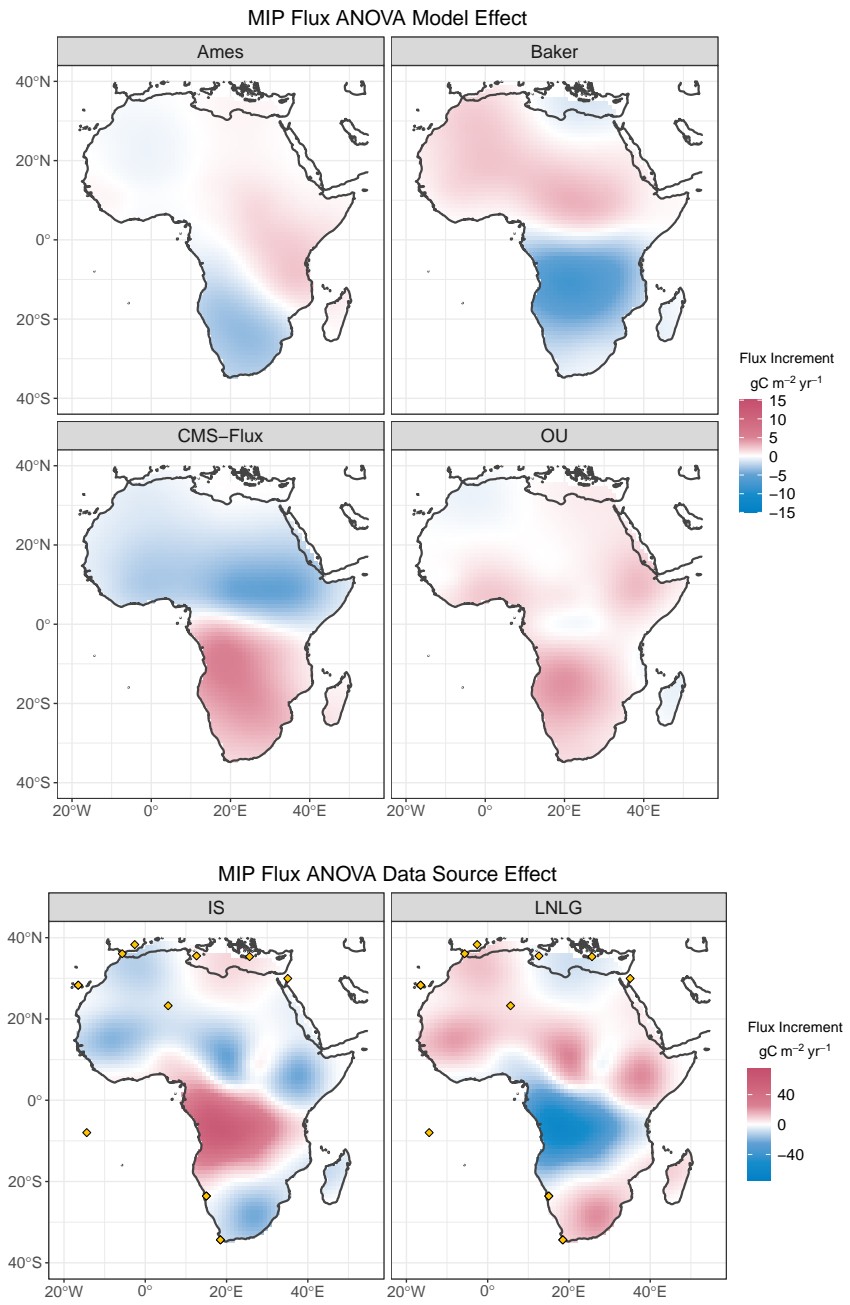

**Figure 10.** Posterior mean for the functional ANOVA main effect for flux model ($\alpha$, top) and for data source ($\beta$, bottom) for JJA 2016 over Africa. In situ observation locations are indicated by gold diamonds.

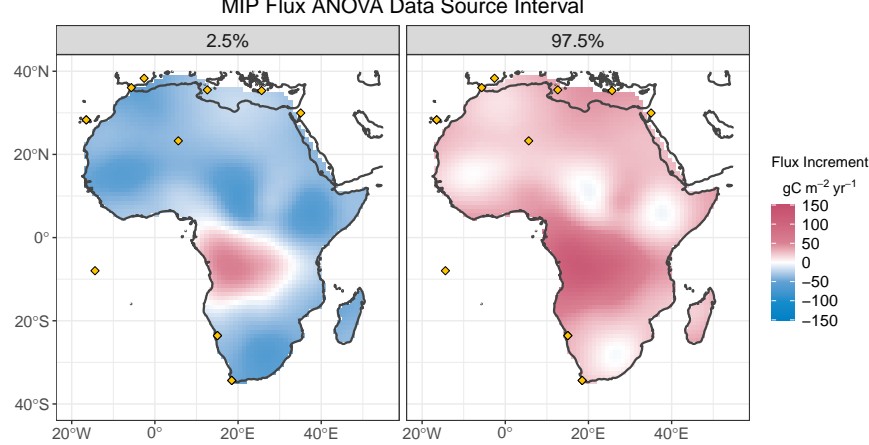

**Figure 11.** Posterior credible intervals for the functional ANOVA data source effect $\beta^*$, the difference between in situ (IS) and OCO-2 land (LNLG) inversions for JJA 2016 over Africa. Left (right) panel depicts the 2.5th (97.5th) percentile of the posterior distribution for each location. In situ observation locations are indicated by gold diamonds.

the relative roles of transport model uncertainty and input data challenges such as bias and incomplete sampling. Over both continents, the range of spatial correlation was largest for the model effect, suggesting that model-to-model implementations contribute to differences at large scales, including aggregated regional fluxes (Peiro et al., 2022; Crowell et al., 2019).

The four inversion systems represented in the MIP functional ANOVA use the same inverse method and have similar spatial resolution in their flux solutions. This subset was selected to illustrate the ANOVA, including the Vecchia approximation for GPs, for a factor with more than two levels, where a more complex set of contrasts is employed to preserve the sum-to-zero constraints. This demonstration indicates that the extension to more than two levels per factor is attainable methodologically and computationally. The estimation could be extended to the full collection of inversion systems in the OCO-2 MIP collection. This extension would modestly increase the computational burden of the MCMC, but the intensive operations on the GP precision matrices would still be executed just once per ANOVA component per MCMC iteration, as noted in the supplement (Section S.1.3.1). MCMC convergence could be somewhat more challenging with more levels per factor.

ANOVA methods and the resulting estimates can be used to devise potentially unequal weights for combining flux estimates into a consensus flux estimate (Cressie et al., 2022). This weighting is employed with univariate ANOVA when there are different variances, e.g. $\sigma_{\alpha,i}^2$, for all levels of a particular factor. Provided these additional parameters can be estimated reliably from the available data, the weights are taken to be inversely proportional to the level-specific variances. In the functional ANOVA setting in this paper, related extensions for the ANOVA GPs are possible. The GP models used in this paper result in a variance that differs by ANOVA component but is constant across space. Alternative parameterizations with heterogeneous standard deviations across space could be developed. Extending the model in this fashion might be feasible for the ANOVA error term in particular, but the challenge would be to formulate a flexible yet parsimonious model for the standard deviations.

This could be achieved by linking the variability to a land cover mask (Villalobos et al., 2020), or scaling the standard deviation to be proportional to the prior flux uncertainty. Kang and Cressie (2013) used a spatial random effects (SRE) model for functional ANOVA components that could result in additional nonstationarity across space. Future research could explore these modifications while aiming to maintain the computational efficiency of the functional ANOVA inference.

The current implementation of functional ANOVA for carbon flux estimates has extended related applications to climate models (e.g. Kaufman and Sain, 2010; Sain et al., 2011; Kang and Cressie, 2013) in a number of ways, including the estimation of ANOVA effects for factors with more than two levels, the use of a repeated-measures temporal correlation structure in the error process, and the incorporation of the Vecchia approximation for GPs. The data structure for the examples in this paper differ in some key ways from the previous climate applications as well. These previous studies used multiple years in a climate simulation as replicates to infer a spatially-varying climate signal and model effects in the presence of interannual variability. In these studies, the number of replicates was sizable, with $n_\epsilon \approx 30$. The carbon flux examples presented here all used $n_\epsilon = 3$, and the ANOVA error term's GP standard deviation, $\sigma_\epsilon$, is relatively large. This low-replication, high-variance scenario often translates to higher uncertainty in the other ANOVA effects and a tendency for them to shrink to the assumed population mean, zero in this case (Gelman, 2005). Even so, significant differences can be inferred in this case.

Since the functional ANOVA model is applied to flux increments, or the difference between posterior and prior fluxes, some of the functional ANOVA results can be challenging to interpret, particularly for the MIP experiments. For the OCO-2 MIP, each modeling group selected its own prior flux and strategy for gridding results to the common output resolution. These choices, along with transport model impacts, all likely contribute to the spatial patterns in model effects inferred through the functional ANOVA. These characteristics could be more explicitly controlled in the design of future multi-model flux inversion studies. For example, multi-model experiments that use a common prior flux across inversion systems would help diagnose the impact of other sources of variability, such as transport model effects.

As the satellite $CO_2$ record, particularly from OCO-2, extends to multiple years, the methodology can be extended to also include replicates across years. In addition to extending the spatial version to other continents and ocean basins, the functional ANOVA approach can additionally be modified to analyze groups of time series (Kaufman and Sain, 2010; Cuevas et al., 2004), and this is a common method of assessment for regional fluxes (Peiro et al., 2022). Finally, this work has developed a statistical model and appropriate handling of pseudo-replication for adjacent points in time. This additional complexity for the error terms in the statistical model incorporates both spatial and temporal correlation. The estimated temporal correlation for the error process is fairly small for the flux inversion examples presented in this work, but this spatio-temporal model provides a flexible extension to the functional ANOVA methodology. Overall, the functional ANOVA methodology offers suitable flexibility for anomaly detection among discrete collections of Earth system models.

*Code and data availability.* The OCO-2 V9 MIP surface gridded fluxes used in the examples are available from https://www.esrl.noaa. gov/gmd/ccgg/OCO2_v9mip. R code for processing the flux fields and implementing the functional ANOVA via MCMC is available at https://github.com/co2anomaly/flux_fanova. The repository release used to produce the examples is available at https://doi.org/10.5281/

zenodo.8248871. The supporting datasets for the examples, including the MCMC posterior samples, are available at https://doi.org/10.5281/zenodo.8152509

*Author contributions.* J.H.: Project administration, Conceptualization, Methodology, Software, Formal analysis, Data curation, Writing – Original Draft, Writing - review & editing, Visualization. M.K.: Conceptualization, Methodology, Software, Writing - original draft, Writing - review & editing. H.N.: Conceptualization, Methodology, Software, Data curation, Writing - review & editing. V.Y.: Project administration, Conceptualization, Methodology, Supervision, Writing - review & editing. J.L.: Conceptualization, Methodology, Formal analysis, Data curation, Writing - review & editing.

*Competing interests.* The authors declare no competing interests.

*Acknowledgements.* The research described in this paper was performed at the Jet Propulsion Laboratory, California Institute of Technology, under contract with NASA. Support was provided by the Making Earth System Data Records for Use in Research Environments (MEaSUREs) program. The authors thank David Baker, Amy Braverman, Noel Cressie, Susan Kulawik, and the OCO-2 flux team for helpful discussions. The authors further appreciate the thoughtful comments from two reviewers on this manuscript.

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
