# Peer review of "Functional ANOVA for Carbon Flux Estimates from Remote Sensing Data"

_Geoscientific Model Development, 2022_

## Referee Comment (RC2)

**Review of GMD-2022-230**

This study applies a functional ANOVA (analysis of variance) method to extract spatially-varying effects related to different experimental factors from carbon dioxide inversion results. The ideas are not completely new, but some computationally efficient tricks are introduced, and the methodology is applied to a few different questions in the context of interpreting fluxes from atmospheric inversions. The goal is to quantify the impact of different sources of data, different methods of data aggregation, and the use of different inversion models. The code has been made public, and appears to be sufficiently documented. (I did not replicate the results, but a cursory inspection of the code repository suggests that this would be possible.) The study is generally well written, and the figures are of high quality. The subject matter is appropriate for publication in GMD once some issues are addressed.

A significant concern about the methodology is related to the use of adjacent months (June, July, and August) as independent replicates to represent the accumulated JJA signal. This seems questionable in two regards:

1. These are not truly independent: There is certainly some correlation between the fluxes of adjacent months, in terms of the data constraint, the geophysical processes involved, and likely even the optimization of the fluxes via a 4DVar approach.
2. These are not realizations of the same quantity: The fluxes for June are not a replicate of the fluxes for August, due to both seasonal cycles (as can be clearly seen in the columns of Figure 1) and differences in short-term anomalies (e.g. June could be anomalously hot and dry while August of the same year could be anomalously cool and wet).

Both of these points call into question whether this choice is valid. This is rather different than the previous studies mentioned in the conclusion, which used different years (not adjacent months!) to infer spatially-varying climate signals and model effects in the presence of interannual variability. (The studies cited that analyzed climate models also used an order of magnitude more replicates.)

The findings seem to suggest that, indeed, these are not appropriate to use as replicates. Lines 242 and 243 point out that the estimated GP standard deviation for the is largest for the error fields (10 times larger, in fact), which indicates that the month-to-month variability within each treatment combination is rather large. Indeed, the standard deviation of the error term is largest for all the experiments, as seen in Tables 3 and 4. This is discussed in the conclusion, with the remark that such a "low-replication, high-variance scenario often translates to higher uncertainty in the other ANOVA effects and a tendency for them to shrink to the assumed population mean", namely zero. Is there a way that the study could have been conceived in order to have a larger number of replicates?

Regardless, the authors argue that they have been able to extract "significant anomalies". It is unclear to me how to judge the significance of these results. Is it just because there are coherent spatial patterns? The 95% credible intervals are generally smaller for the error than for the mean, $\alpha$, $\beta$, or interaction terms – does this say something about the relative significance of the results? The authors argue that the large standard deviation of the error fields is offset by the relatively small estimated range parameter of the error term. How is this distance to be interpreted? In some cases, the range parameter is significantly smaller than the size of the models' pixels (e.g. a median value of 70.5 km for the range parameter of the error for the African region in Table 4, compared to model resolutions an order of magnitude larger, from 4°x5° to 6.7°x6.7°, based on Table 1 of Peiro et al., 2022). It would be helpful to provide some more information about how to interpret these numbers. As another example, the unitless smoothing parameter is reported in the tables, but these results are never discussed in the text. What do these mean?

I think it would be useful to include some discussion about the similarities and the differences between your approach (and results) and those of Cressie et al. (2022). The latter study applied ANOVA to the same two years of very similar OCO-2 MIP results (a previous iteration), also using individual months of a three-month season as replicates (which seems questionable in their study as well). The application was different, however, as they were using it to develop an optimally-weighted multi-model mean for regional fluxes, and considered all seasons and regions covering the whole globe.

This raises the question: why did you decide to focus on JJA, and why only on this one season? How would you expect the results to change if different seasons were analyzed? Why did you only use two years of data, when the OCO-2 MIP results you are using (from Peiro et al., 2022) provide four full years of valid results? What led you to look only at these specific regions? There is some discussion of why North America and Africa were chosen to illustrate the technique, due to contrasting measurement coverage, but there is no clear argument as to why Eurasia was chosen for the first part of the study. Providing a justification for why a given region was chosen to explore the specific question would make the scientific results more compelling.

In general, more scientific interpretation of the spatial patterns that are presented as results would improve the paper. In particular: in the plots of the "Data Source Effect" in Figures 7 and 10, how do these patterns correspond to the location of in situ measurement sites? (Yes, the stations are shown in Figure 1 of Peira et al. (2022), but putting the site locations onto your figures would make this easier to interpret.) Where is there persistent cloud cover and/or good retrievals of OCO-2 data in North America and Africa during the period in question? Interpreting the structure of these "Data Source Effect" figures with reference to the measurements would be helpful.

Another concern regarding the methodology is related to the fact that all the ANOVA analysis in the MIP portion of the study is done on the flux increment, i.e. the posterior flux minus the prior flux for a given space and time, as shown in Equation 1. I believe that this is problematic, because the four models whose results are compared do not have identical priors. Extracted from Table 1 of Peiro et al. (2022), the following prior fluxes were used:

| Simulation name | Prior land | Prior ocean | Prior fire |
| --- | --- | --- | --- |
| Ames | CASA-GFED4.1s | CT2019OI | GFED4.1s |
| CMS-Flux | CARDAMOM | ECCOS-Darwin | GFED4.1s |
| OU | CASA-GFED3 | Takahashi | GFEDv3 |
| Baker | CASA-GFED3 | Takahashi | GFEDv3 |

All the inversions are constrained by the same data, and we would expect that if transport is perfect, the data constraint is sufficient, and errors are well-characterized, the four models should converge to the same posterior fluxes. That does not mean that their flux *increments* will be the same. Consider a case where the prior of model A is biased systematically high over a given region, while the prior of model B is biased systematically low. Both models converge to the same response, but when analyzing the increment ($y_{ijk}(\mathbf{s}) - y_{ijk(0)}(\mathbf{s})$), model A's would be negative and model B's would be positive. If the biases were equally biased, the "consensus flux anomaly" of this ensemble of two would be zero. This is written in the text, e.g. in lines 283-284: "Noting again that the analysis is carried out the inversions' deviations from their respective priors, the map depicts a consensus flux anomaly from the collection of inversions analyzed." How can we interpret a "consensus flux anomaly" when the models are starting from different priors? (This does not apply to the first part of the study, only the MIP results.)

Going back to the theoretical model A and model B from before: these increments would presumably show up as negative and positive flux model differences with a large spatial range, perhaps similar to the all-red (Ames) and all-blue (CMS-Flux) posterior mean maps seen for α for JJJA in North America (Figure 7). However, the authors attribute this to "large-scale regional flux differences among inversion systems with different driving atmospheric transport" (L275-276). This could be the case, but without having common priors, the effects of different priors and different transport cannot be teased apart using this method. Ideally, such an analysis would be carried out on inversion results with identical priors, so that we could truly say something about the influence of the transport model and optimization. Lacking that, this limitation needs to be made completely clear, so that the reader is not misled.

As a more minor/technical note: The treatment of the different spatial resolutions in the MIP part of the study should be explicitly explained. For the first part of the study, only results from the CMS-Flux model are used, and the plots appear to match the 4°x5° resolution of the model. For the MIP, models of different resolutions are compared (but none with a higher resolution than the still relatively coarse 4°x5° resolution of CMS-Flux), but the data appear to have been resampled onto a higher spatial grid in Figures 6, 8 and 9. In Figures 7, 10 and 11, the resolution appears smoother still. What procedure was followed?

**Minor/typographical comments:**

L1-2: This sentence is a bit odd, making it sound as if measurements made today (present tense) span several years. Perhaps: "has now produced atmospheric greenhouse gas concentration estimates covering a period of several years"?

L8: Not clear what is meant by "mode differences" here.

L10: For inversions over different continents, or for fluxes over different continents?

L17: The language here makes it sound as if the satellites are providing an estimate of global CO2 concentrations frequently, which is not the case. Instead of "frequent", perhaps quantify it? How many good quality measurements per year, on average?

L18: Has flux inversion become demonstrably more tractable (not quite sure what that means here) and precise due to satellite measurements? If so, please provide a reference providing evidence of this!

L24-25: Not clear what is meant by "dependence can be exploited in quantifying uncertainty". Dependence of what on what? Please explain.

L39: collection Earth -> "collection of Earth"

L41: Hyphenate "multiple-component" as a compound adjective, also "spatially-aggregated" on L45.

L47: with associated -> "with an associated"

L55: Is the year missing from Sain et al. reference here? Is this consistent with the Copernicus style guide?

L56: Would replace the comma with a period here, breaking into two sentences. (Or at least a semicolon.)

L57: Would be easier to parse if "used in the current work" were preceded and followed by a comma.

L70: Which version of ACOS?

L73-75: Please clarify about the creation of super-observations through aggregation. How it's written here, it sounds as if the data are aggregated across the whole orbit. Aggregation to super-observations is not the same as what is generally considered a L3 product, which is rather gap-filled concentration fields, perhaps through data assimilation to optimize atmospheric state (instead of fluxes). This is explained further in 2.1, but seems inconsistent with what is written here.

L101: What is SSDF? I believe this has not been defined.

L103: variables of interest?

L120: Perhaps specify that you mean ACOS version 9?

Caption Figure 2: remove "based"

L181: and: should maybe be changed to ", which" or "that"?

L224: parameters -> parameters'

L225: are -> is

L227: is -> are

L231: is -> was

L232: The wording here is a bit dense. Perhaps better: "which makes the MCMC algorithm more computationally efficient."

Caption Table 3: Not clear what "in the records of [the] fused CO2 experiment" means here. Perhaps just "for the fused CO2 experiment"? Also, "of 95% posterior credible intervals" -> "of the 95% credible intervals of the posterior". The same comment applies to the caption of Table 4 as well. In general, I was a bit confused by the use of "the records of fused CO2 experiment", also in the captions of Figures 4 and 5.

Figure 4: Units are missing. Somewhere in the caption it should be made clear that the "mean" is actually the mean difference from the prior for JJA over 2015-2016, rather than the mean flux itself. The label on the lower right panel should be "Interaction" rather than "Interact".

L270: Differences in behaviour, or differences in data coverage?

L283: Missing the word "on" (carried out on the inversions'…)?

L315-316: Really? The probability that the difference across years was larger than the aggregation method effect did not exceed 0.6. This does not convince me that using such a fused product provides internally consistent information for analyzing e.g. interannual variability in fluxes. Perhaps I have misunderstood something.

L339-340: Is the word "variance" (or something else) missing? i.e. "Alternative parameterizations with heterogeneous *variance* across space could be developed."

**Supplement:**

L32: model -> models

**References**

Cressie, N., Bertolacci, M., and Zammit-Mangion, A.: From Many to One: Consensus Inference in a MIP, Geophys. Res. Lett., 49, e2022GL098 277, https://doi.org/10.1029/2022GL098277, 2022.

Peiro, H., Crowell, S., Schuh, A., Baker, D. F., O'Dell, C., Jacobson, A. R., Chevallier, F., Liu, J., Eldering, A., Crisp, D., Deng, F., Weir, B., Basu, S., Johnson, M. S., Philip, S., and Baker, I.: Four years of global carbon cycle observed from OCO-2 version 9 and in situ data, and comparison to OCO-2 v7, Atmos. Chem. Phys., 22, 1097–1130, https://doi.org/10.5194/acp-22-1097-2022, 2022.

---

## Author Comment (AC1)

**Manuscript: Functional ANOVA for Carbon Flux Estimates from Remote Sensing Data**

J. Hobbs, M. Katzfuss, H. Nguyen, V. Yadav, J. Liu

August 2023

**1 Responses to Reviewer 1 (RC1)**

**1.1 Major comments**

- **I'm concerned that one of the assumptions about the error distribution, namely that the realisations are independent, may not be valid. The realisations are taken as successive months within a three-month block. In my experience, geophysical fields and their associated error distributions are typically correlated in space and time. The authors need to address this issue directly, perhaps demonstrating numerically that the residual fields cannot be distinguished from random draws, or arguing persuasively why this should be overlooked for the sake of illustrative example. One thing to consider here, and probably should be raised in the revision, was whether the inversions were conducted as a "long" run (i.e., at least spanning all of the three months considered), or whether they were run separately for the individual months in question.**

  This is indeed an important point regarding the functional ANOVA model used in the first version of the manuscript. As the reviewer notes, all inversions are in fact performed as "long runs", considering the entire time record, so temporal correlation is possible. Upon receiving similar comments from both reviewers, we revisited the methodology and have modified the functional ANOVA model to allow for temporal correlation in the error terms $\epsilon_{ijk}(\mathbf{s})$. The modified statistical model is outlined in the revised manuscript in Section 3.1. The resulting estimates for the temporal correlation parameters are discussed with the other results in Section 4. The resulting estimates turn out to be relatively small but somewhat important for the overall inference, and the magnitude of the correlation is an interesting aspect of the analysis in its own right.

  The updated supplement includes some additional details on the specification of the temporal correlation structure and its integration into the MCMC algorithm. With the manuscript revision, we include a new release of the software repository that includes the spatio-temporal model for the error process.

- **It is unclear what is going on with the ocean areas. The distribution of fluxes is very different for land areas or oceans. Similarly, the distribution of fluxes depends very much on the type of vegetation cover. Consider for a moment differences between tropical rainforest vs savannah vs deserts vs high-latitude boreal forests. Are we assuming a uniform standard deviation? This should be discussed.**

  The analysis does not include fluxes over the oceans, and we provide additional detail on this in the revised manuscript and address in detail in the response to the next comment below.

  The additional remark on land/vegetation cover is an important comment. As currently implemented, the functional ANOVA Gaussian process models assume a constant variance for each of the components, including the error process. This combination does offer substantial flexibility to fit the

[Figure]

Figure RC.1: Estimated error standard deviation $\hat{\sigma}_\epsilon$ for location-specific ANOVA model fits for the MIP Africa example. Units are gC m$^{-2}$ yr$^{-1}$

spatio-temporal variability in the flux fields. Even so, processes could exhibit spatially heterogenous variability. We have conducted some exploratory analysis to address the potential for heterogeneous variances, particularly for the ANOVA error term. One tool for this investigation is location-specific ANOVA model fits. These individual results provide an estimate of the error standard deviation $\hat{\sigma}_\epsilon$. These estimates are displayed in Figure RC.1.

This analysis does suggest that the error process variability could change across the region. While the magnitude can apparently be connected to land cover, other factors, including algorithmic choices for the various flux inversion systems likely contribute to this result as well. The prior flux uncertainty could contribute to this behavior as well, but that information is unavailable in the OCO-2 flux MIP output. At this juncture, we do not have a parsimonious and identifiable representation in the model for this heterogeneity. In the revised manuscript, the discussion includes additional remarks on this topic.

- **Also on the topic of oceans, it seems that the fluxes for most of the oceans are zero, or close to zero on the colour scales shown, except for some small areas (e.g., over the Mediterranean or Red Seas in Figure 10). Is this intentional? Is a land-mask being imposed? I recommend discussing this further.**

Yes, a mask has been imposed, and all of the analyses are for fluxes over land regions only. Each example uses fluxes over specified TransCom regions, as outlined in Table 2. For additional exposition, we have added some discussion of the selected TransCom regions at the end of Section 2.1 and Section 2.2 in the revised manuscript.

**1.2 Minor Comments**

- **Multiple places: There are multiple references to the supplement of this manuscript. I recommend making the references more specific, such as to particular figures, tables, sections, pages, etc.**

This is an excellent suggestion. For each reference to the supplement in the revised main manuscript, we have included a relevant specific section or subsection.

- **Abstract: I thought this was rather short. I recommend adding a few sentences expanding upon conclusions and implications of this work.**

  The revised manuscript includes an updated abstract. This abstract includes additional discussion of the results, particularly the contrasts among the spatio-temporal covariance parameters for the ANOVA components. The residual term exhibits similar behavior across all of the examples, and this is also mentioned in the revised abstract.

- **L3: "are combined" → "can be combined"**

  This change has been made in the revised manuscript.

- **L4: "impact flux estimates" → "impact such flux estimates"**

  This change has been made in the revised manuscript.

- **L8: I found the term "mode differences" somewhat cryptic. Please review and reformulate.**

  The term has been removed from the updated abstract to provide additional clarity on the results of the CMS-Flux example over Eurasia.

- **L31: The term "data fusion" isn't particularly well defined. I suggest adding some clarifying remarks or using a different term**

  We agree that a definition of data fusion is needed. This has been added at the above location in the revised manuscript. The definition emphasizes addressing spatio-temporal correlation in producing a common estimate at regular resolution/spacing.

- **L37: "ANOVA methodology" → "The ANOVA methodology"**

  This change has been made in the revised manuscript.

- **L41: "multiple component" → "multi-component"**

  This change has been made in the revised manuscript.

- **L47-48: Is the method of Cressie et al. (2022) based on functional ANOVA? If so, I suggest noting this.**

  Cressie, Bertolacci, and Zammit-Mangion (2022) use the classical univariate ANOVA approach for spatially-aggregated flux estimates. In that work, the ANOVA model provides information on partitioning of variability among factors such as the inversion model, but it does not provide any information on the extent of spatial correlation. Therefore we think the current paper provides a meaningful contribution to the flux inversion literature with these additional results using functional ANOVA.

  We appreciate both reviewers raising the importance of framing the functional ANOVA approach and results in the context of the recent contribution from Cressie et al. (2022). In the revised manuscript, we have added a sentence at the above location in the Introduction to highlight our contribution. Further, the connection to Cressie et al. is highlighted again in the first two paragraphs of the Conclusions.

- **Page 2: Reading this far, I was unsure whether the authors were talking about gridded data or not. This became clear later on, but I suggest making a note about this somewhere around this point in the Introduction.**

  This is a good observation. While the statistical model (i.e. Gaussian process) is suitably general for irregular spatial data, the references cited and our work are implemented on gridded datasets. This has been added to the Introduction in the revised manuscript.

- **Page 2: I was left wondering how functional ANOVA handles different spatiotemporal resolutions (or, more generally, different spatiotemporal partitioning). Do all of the data need to be on a common grid?**

  With each of the examples in our study, the spatial resolutions are the same for all factor combinations, but the resolution is different between the MIP examples and the CMS-Flux aggregation example. More generally, functional ANOVA could operate on data with different support/resolution, but an additional change-of-support component would likely be needed in the model.

- **L63-64: References to Sections 2 and 3 are swapped around. Review.**

  The section references have been corrected in the revised manuscript.

- **L69: I suggest noting the units of $XCO_2$**

  The units (parts per million) have been noted at this spot in the revised manuscript.

- **L70: Do you ave a version number for the ACOS retrieval algorithm?**

  The version number (V9) has been added to the revised manuscript.

- **L73: "within a single polar orbit" - do you mean "within a single model grid-cell"?**

  This phrasing was intended to convey that only retrievals in the same orbit are aggregated together, but the spatial resolution is the more important aspect. This has been changed in the revised manuscript.

- **L75: "a precise" → "a more precise"**

  This change has been made in the revised manuscript.

- **L83: I find URLs within the text, particularly when it is messy such as DOIs, rather unsightly. I suggest creating a bibliography entry for this dataset, and shifting the URL there.**

  We appreciate the suggestion. The dataset has been added as a reference, with the DOI provided in the reference list.

- **L101: "SSDF" - Has this acronym been introduced?**

  We intended to drop the SSDF acronym from the manuscript. The revised manuscript has removed this item.

- **L103: "The variable of interest" → "The variables of interest"**

  This change has been made in the revised manuscript.

- **L122: "the individual retrievals are both variable and moderately correlated in space and time" → "the individual retrievals are both uncertain and the associated errors are moderately correlated in space and time"**

  The suggested modification conveys our intended message effectively and has been incorporated into the revised manuscript.

- **L122: What is meant by "moderately" here?**

  We have added an additional sentence here to mention that empirical analysis of spatial correlation for OCO-2 XCO2 retrievals has found modest (i.e. $< 0.4$) among neighboring retrievals, plus correlation range estimates on the order of 5-10 km.

- **Figure 1 caption: I don't think the acronym "JJA" has been introduced until now.**

  The acronym is defined in the text, but we have added it to the caption as suggested.

- **L123: "spatially aggregated" - as a gridded or regional average?**

  These are gridded averages. We have noted the approximate along-track extend (75 km) in the revised manuscript.

- **L129: "have similar resolution" → "have similar spatial resolution"**

  This change has been made in the revised manuscript.

- **Figure 2 caption: "over North America." → "over North America, with columns for the different inversions."**

  We have incorporated this additional clarification into the revised figure caption.

- **Table 1: I recommend adding an extra column with a key reference for each retrieval or each retrieval dataset.**

  This additional column has been added in the revised manuscript. In addition, Peiro et al. (2022) and Crowell et al. (2019) contain supporting materials with these references.

- **L166-167: Here we see the issue of months within a season being used for different replicates (see the first of my "Major comments" above).**

  As noted in the response above, the methodology has been modified to incorporate temporal dependence in the error process. We appreciate both reviewers' remarks on this important aspect of the methodology.

- **L170: "are often" - why not "must be"? Is there an alternative?**

  This change has been made in the revised manuscript.

- **L172: "the factor and interaction effects add to zero, e.g." → "the factor effects add to zero, i.e."**

  This change has been made in the revised manuscript.

- **Equations below L172, and L175: Why not put these on the same line? Would they not fit?**

  The equations have been combined into a single line in the revised manuscript.

- **L195: Here the realizations are assumed to be independent (see the first of my "Major comments" above).**

  The revised model with temporal autocorrelation in the error process is discussed at this point in the revised manuscript.

- **L214: $\theta$ was not defined, as far as I could see.**

  The revised manuscript includes a definition of $\theta$ as the full collection of spatial and temporal covariance parameters at this point in the paper.

- **L215: What do you mean by "diffuse"?**

  We have changed the wording to "high variance" to reflect substantial a priori uncertainty.

- **L223: "for the component and the noise" → "for the components and the noise"**

  We intended this to be "component being updated".

- **Page 11: Here I was hoping to see more about the question of a land mask, or different distributions (e.g., heteroscedastic) for land/ocean areas.**

  Additional discussion of the (land-only) TransCom regions studied has been included in Section 2. Only fluxes over these land masks are analyzed.

- **Page 12: It gets mentioned much later in the Results section (L285) that negative anomalies correspond to carbon uptake by the land, and positive anomalies to the release of carbon into the atmosphere. I think this is an important part of the interpretation of the results shown, and as such suggest introducing this much earlier in the Results section (e.g., within the first few paragraphs, maybe also in the caption for Figure 4).**

We appreciate this suggestion, as the conventions for fluxes will not be known to all readers. In the revised manuscript, we have decided to introduce this convention for the sign of the fluxes when the datasets are presented in Section 2. We have also added further discussion of the signs of the ANOVA components at the beginning of the results section. The use of the deviations from the prior in the analysis adds some subtlety to the interpretation, and we have emphasized this point at the beginning of the Results section, in addition to the impacts for specific results.

- **L240-245: I would have like to see a bit more interpretation. For example, what are the implications of the estimates for the range parameter being an order of magnitude larger for the mean and year effect than the other terms? What does it mean that the error standard deviation is so much larger than that of the other terms? What can one read from the relatively narrower credible intervals for the error standard deviation than those of the other terms?**

The revised manuscript includes additional discussion of these results at the beginning of Section 4.1. In particular, we emphasize that the mean and year effects are connected to the behavior of the carbon cycle across the JJA season, and the larger range estimates indicate that the functional ANOVA has estimated spatially coherent perturbations in the carbon cycle over this region. The other terms in the functional ANOVA represent data processing and algorithm characteristics, and they exhibit limited spatial coherenece.

The width of the posterior credible intervals for the covariance parameters is primarily a function of the number of spatial fields that inform the estimates. This is analogous to degrees of freedom in traditional ANOVA. The availability of the pseudo-replicates yields $n_\alpha \ n_\beta n_\epsilon$ fields to inform the covariance parameters for the error process. On the other hand, $n_\alpha - 1$ fields inform the parameters for $\alpha^*$, and $n_\beta - 1$ fields inform the parameters for $\beta^*$.

- **L254: "The left panel shows $Pr(|\mu(\mathbf{s})| > |\beta^*(\mathbf{s})|)$" - How was that estimated? With reference to the posterior samples?**

Yes, these probabilities can be estimated from the Markov chain Monte Carlo samples for both processes. We have mentioned this in the text of the revised manuscript.

- **L260: It would be worth noting in the text that the posterior probabilities for the secondary question, $Pr(|\alpha^*(\mathbf{s})| > |\beta^*(\mathbf{s})|)$, are clearly lower than for the primary question, being $Pr(|\mu(\mathbf{s})| > |\beta^*(\mathbf{s})|)$.**

We have included additional discussion of these combined results in the revised manuscript to address this comment and related comments from the second reviewer.

- **L285: "negative anomalies (increased uptake)" - as noted above, this should be mentioned earlier.**

In the revised manuscript, we have introduced this convention for the sign of the fluxes when the datasets are presented in Section 2.

- **Figures 7 & 10, bottom panels (data source effect): I would have been interested to see the impact of the locations of the in-situ sites. Consider adding dots for the locations of the CO2 monitoring stations.**

Both reviewers offered a similar suggestion, and we have added in-situ locations to Figures 7, 10, and 11 in the revised manuscript. The implications of in-situ density are discussed for the Africa example, in particular.

- **L295: "The prior fluxes over the continent (not shown)" - Why not put this in the supplement?**

  This is a good suggestion. We have included maps of the prior fluxes for all examples in the revised supplement.

- **Around L293-294: I would suggest noting that for much of the domain, the direction of the flux estimated by the posterior is ambiguous.**

  This note has been added to the results discussion in the revised manuscript. The paragraph in question has been modified further based on the comments from the other reviewer.

- **Last paragraph on Page 15: I would recommend expanding some of the interpretation. On the question of the data sources, adding to the Figure 10 the locations of the monitoring sites could provide some basis for some of the interpretation of the data source effect.**

  The revised manuscript has included some additional interpretation for these results, particularly for the data source effect and the locations of in situ observations. The locations of observation sites have been added to Figure 10.

- **L320: "The functional ANOVA identified local consensus flux anomalies for both continents in the presence of variability across inversion systems and atmospheric CO2 data sources." → "The functional ANOVA identified local consensus in flux anomalies for both continents across different inversion systems and atmospheric CO2 data sources."**

  This change has been made in the revised manuscript.

- **L339: "heterogeneous across space" - I think there's a word missing here. Do you mean heterogeneous standard deviations?**

  Yes, this has been corrected in the revised manuscript.

- **Discussion: I think it would be useful to expand upon the importance of spatial heterogeneity. See the second of my Major comments.**

  This is indeed an important point. In addition to the previous responses, we have added further discussion in the Conclusion to the revised manuscript. This added discussion includes the potential for incorporating land cover information and modeling the error process variance as a function of the prior flux uncertainty. This information is not available in the OCO-2 flux MIP output, but would potentially be provided from flux inversions. A final point is that any modified parameterization would have to be considered carefully in order to maintain identifiability and computational tractability in the inference.

- **References, multiple places: CO2 should be $CO_2$ and XCO2 should be $XCO_2$**

  We appreciate the careful reading of the references. The symbols have been modified for consistency in the revised manuscript.

- **References, multiple places: at least two of the author lists are truncated (finishing with "et al.") while others extend to many authors. Please make this consistent, preferably checking the GMD Guide to Authors.**

  The references have been modified to include the complete author lists, as noted in the GMD author guide.

**References**

Cressie, N., Bertolacci, M., & Zammit-Mangion, A. (2022). From many to one: Consensus inference in a MIP. *Geophys. Res. Lett.*, *49*, e2022GL098277. doi: 10.1029/2022GL098277

Crowell, S., Baker, D., Schuh, A., Basu, S., Jacobson, A. R., Chevallier, F., ... Jones, D. B. A. (2019). The 2015-2016 carbon cycle as seen from OCO-2 and the global in situ network. *Atmos. Chem. Phys.*, *19*, 9797–9831. doi: 10.5194/acp-19-9797-201

Peiro, H., Crowell, S., Schuh, A., Baker, D. F., O'Dell, C., Jacobson, A. R., ... Baker, I. (2022). Four years of global carbon cycle observed from OCO-2 version 9 and in situ data, and comparison to OCO-2 v7. *Atmos. Chem. Phys.*, *22*, 1097–1130. doi: 10.5194/acp-22-1097-2022

---

## Author Comment (AC2)

**Manuscript: Functional ANOVA for Carbon Flux Estimates from Remote Sensing Data**

J. Hobbs, M. Katzfuss, H. Nguyen, V. Yadav, J. Liu

August 2023

**2  Responses to Reviewer 2 (RC2)**

**2.1  Major comments**

- **This study applies a functional ANOVA (analysis of variance) method to extract spatially-varying effects related to different experimental factors from carbon dioxide inversion results. The ideas are not completely new, but some computationally efficient tricks are introduced, and the methodology is applied to a few different questions in the context of interpreting fluxes from atmospheric inversions. The goal is to quantify the impact of different sources of data, different methods of data aggregation, and the use of different inversion models. The code has been made public, and appears to be sufficiently documented. (I did not replicate the results, but a cursory inspection of the code repository suggests that this would be possible.) The study is generally well written, and the figures are of high quality. The subject matter is appropriate for publication in GMD once some issues are addressed.**

  We appreciate the reviewer's assessment of the work and its context within the existing literature. The insightful comments have motivated revisions in our approach and presentation in this manuscript.

- **A significant concern about the methodology is related to the use of adjacent months (June, July, and August) as independent replicates to represent the accumulated JJA signal. This seems questionable in two regards:**

  1. **These are not truly independent: There is certainly some correlation between the fluxes of adjacent months, in terms of the data constraint, the geophysical processes involved, and likely even the optimization of the fluxes via a 4DVar approach.**

     This is indeed an important point regarding the functional ANOVA model used in the first version of the manuscript, and we have revisited the methodology after receiving similar comments from both reviewers. The revised functional ANOVA model is defined in Section 3.1 and allows for temporal correlation in the error terms $\epsilon_{ijk}(\mathbf{s})$. The resulting estimates for the temporal correlation parameters are discussed with the other results in Section 4. It should also be noted that this temporal correlation is for the error/residual after accounting for the remaining ANOVA terms, which are constant in time and vary across space. The estimated temporal correlation parameters are relatively small but impact the inference for the ANOVA terms to some degree.

  2. **These are not realizations of the same quantity: The fluxes for June are not a replicate of the fluxes for August, due to both seasonal cycles (as can be clearly seen in the columns of Figure 1) and differences in short-term anomalies (e.g. June could be anomalously hot and dry while August of the same year could be anomalously cool and wet).**

This point may speak to a benefit in using the flux increments in the functional ANOVA. This approach accounts for some of the seasonal evolution that is also present in the prior fluxes. This choice represents a compromise that facilitates analysis of the intraseasonal variability to identify some coherent spatial patterns, which are present for some of the remaining ANOVA components (overall mean and main effects) for the examples. Indeed short-term anomalies within a season can be present and these would be realized in the error process. The relatively small magnitudes of spatio-temporal correlation in the error process parameters would suggest that any large-scale short-term anomalies might be dominated by other sources of variation in the residual.

**Both of these points call into question whether this choice is valid. This is rather different than the previous studies mentioned in the conclusion, which used different years (not adjacent months!) to infer spatially-varying climate signals and model effects in the presence of interannual variability. (The studies cited that analyzed climate models also used an order of magnitude more replicates.)**

The modification of the model to include temporal correlation for adjacent months, along with use of flux deviations from the assumed prior, help address these important challenges in the examples presented. The reviewer notes the important distinction in defining pseudo-replicates in the current study versus other instances in the literature. We have highlighted this distinction in the revised manuscript's discussion of the functional ANOVA model in Section 3.1.

- **The findings seem to suggest that, indeed, these are not appropriate to use as replicates. Lines 242 and 243 point out that the estimated GP standard deviation for the is largest for the error fields (10 times larger, in fact), which indicates that the month-to-month variability within each treatment combination is rather large. Indeed, the standard deviation of the error term is largest for all the experiments, as seen in Tables 3 and 4. This is discussed in the conclusion, with the remark that such a "low-replication, high-variance scenario often translates to higher uncertainty in the other ANOVA effects and a tendency for them to shrink to the assumed population mean", namely zero. Is there a way that the study could have been conceived in order to have a larger number of replicates?**

The large variance of the error process is indeed a common characteristic across all of the examples presented. We note that this is a result of not only high month-to-month variation, but also spatial variability at local scales after accounting for the other ANOVA effects. This does not necessarily invalidate the methodology in a strict sense, rather it is a characteristic of the data, and it is not an uncommon occurrence in classical ANOVA. The functional ANOVA is able to borrow strength via spatial correlation to some extent, but the large error standard deviation is a challenge for the power to detect meaningful signals for the remaining factors. The comment about alternative formulations is appreciated. For the OCO-2 MIP, additional analysis could make use of the multiple years in the datasets. Given the results from the CMS-Flux inversions for the fused product study, the ANOVA would likely need to consider a factor for interannual variability. This additional model complexity is possible, but we think would detract from the existing messages in the analysis in this paper.

- **Regardless, the authors argue that they have been able to extract "significant anomalies". It is unclear to me how to judge the significance of these results. Is it just because there are coherent spatial patterns? The 95% credible intervals are generally smaller for the error than for the mean, $\alpha$, $\beta$, or interaction terms – does this say something about the relative significance of the results?**

This remark underscores the importance in communicating multiple aspects of the functional ANOVA results, which we seek to improve in the revised manuscript. In the revised Results section, we have added a paragraph that outlines the two broad categories of quantities we examine from the functional

ANOVA inference. The first category is the covariance parameters reported in Tables 3-4. Since these parameters summarize the collective spatial and spatio-temporal variability of each factor, it is useful to contrast their values, and the implications for their spatial coherence, across the factors, as discussed in the Results. In fact, the capability of the method to partition the spatial coherence of the various factors is a key capability. While noteworthy, the actual width of the credible intervals is largely a function of the number of spatial fields that inform the estimates. This is analogous to degrees of freedom in traditional ANOVA. All of the pseudo-replicate monthly fields inform the error process parameters, so they are estimated precisely relative to the $\alpha$, $\beta$, and interaction parameters.

The second collection of results is the set of maps of the ANOVA components. In cases where the pointwise intervals are shown (Figures 8 and 11), areas with intervals that do not contain zero represent a significant anomaly/deviation.

- **The authors argue that the large standard deviation of the error fields is offset by the relatively small estimated range parameter of the error term. How is this distance to be interpreted? In some cases, the range parameter is significantly smaller than the size of the models' pixels (e.g. a median value of 70.5 km for the range parameter of the error for the African region in Table 4, compared to model resolutions an order of magnitude larger, from $4° \times 5°$ to $6.7° \times 6.7°$, based on Table 1 of Peiro et al., 2022). It would be helpful to provide some more information about how to interpret these numbers. As another example, the unitless smoothing parameter is reported in the tables, but these results are never discussed in the text. What do these mean?**

There are two aspects of this comment that we have attempted to address in the revised manuscript. The first has to do with the resolution of the MIP flux estimates. The modeling teams all provided gridded fluxes at $1° \times 1°$, and these are the datasets provided publicly and analyzed in this study.

The second aspect we address is the interpretation of the covariance parameters. The Matérn range and smoothness parameters combine to characterize how correlation between spatial locations decays with separation distance. For an exponential model with smoothness $\nu = 0.5$, the range parameter is the distance at which the correlation reaches $1/e$. For larger values of $\nu$, the correlations decay more slowly at shorter distances. Theoretically, the smoothness identifies the degree of differentiability of the Gaussian process.

We have added some of this practical interpretation to the revised manuscript when the Matérn covariance is introduced in Section 3.1.

- **I think it would be useful to include some discussion about the similarities and the differences between your approach (and results) and those of Cressie et al. (2022). The latter study applied ANOVA to the same two years of very similar OCO-2 MIP results (a previous iteration), also using individual months of a three-month season as replicates (which seems questionable in their study as well). The application was different, however, as they were using it to develop an optimally-weighted multi-model mean for regional fluxes, and considered all seasons and regions covering the whole globe.**

We appreciate both reviewers raising the importance of framing the functional ANOVA approach and results in the context of the recent contribution from Cressie, Bertolacci, and Zammit-Mangion (2022). In the revised manuscript, we have added a sentence at the above location in the Introduction to highlight our contribution. Further, the connection to Cressie et al. is highlighted again in the first two paragraphs of the Conclusions.

- **This raises the question: why did you decide to focus on JJA, and why only on this one season? How would you expect the results to change if different seasons were analyzed? Why did you only use two years of data, when the OCO-2 MIP results you are using**

(from Peiro et al., 2022) provide four full years of valid results? What led you to look only at these specific regions? There is some discussion of why North America and Africa were chosen to illustrate the technique, due to contrasting measurement coverage, but there is no clear argument as to why Eurasia was chosen for the first part of the study. Providing a justification for why a given region was chosen to explore the specific question would make the scientific results more compelling.

In selecting the use cases for demonstrating the functional ANOVA, we sought a balance among multiple objectives for the study. These included motivating carbon cycle questions from the OCO-2 MIP and other recent investigations, motivating questions about sensitivity to data sources/algorithms from the MEaSUREs data fusion effort, and a focus on a parsimonious presentation for a GMD manuscript on assessment of models. This latter objective, along with an interest in the spatial modeling aspect of the functional ANOVA, led to the focus on a single season across different continents. In addition, JJA is a period with recent impactful carbon cycle perturbations and yields sizable uncertainty in MIP consensus flux estimates (Cressie et al., 2022). For the OCO-2 MIP, the interannual variability aspect could be investigated, but this would likely require an additional ANOVA factor. This is feasible but we have kept the analysis to two factors for (relative) simplicity of illustration. We have added further discussion of the carbon cycle science and algorithmic motivation for these choices in the descriptions of the datasets in Section 2 of the revised manuscript.

- **In general, more scientific interpretation of the spatial patterns that are presented as results would improve the paper. In particular: in the plots of the "Data Source Effect" in Figures 7 and 10, how do these patterns correspond to the location of in situ measurement sites? (Yes, the stations are shown in Figure 1 of Peiro et al. (2022), but putting the site locations onto your figures would make this easier to interpret.) Where is there persistent cloud cover and/or good retrievals of OCO-2 data in North America and Africa during the period in question? Interpreting the structure of these "Data Source Effect" figures with reference to the measurements would be helpful.**

We appreciate the suggestions and have implemented several of the points above. We have added in situ locations to Figures 7, 10, and 11 in the revised manuscript. The discussion of the data source differences for Africa includes a mention of reduced data density for OCO-2 over the same area.

- **Another concern regarding the methodology is related to the fact that all the ANOVA analysis in the MIP portion of the study is done on the flux increment, i.e. the posterior flux minus the prior flux for a given space and time, as shown in Equation 1. I believe that this is problematic, because the four models whose results are compared do not have identical priors. All the inversions are constrained by the same data, and we would expect that if transport is perfect, the data constraint is sufficient, and errors are well-characterized, the four models should converge to the same posterior fluxes. That does not mean that their flux increments will be the same. Consider a case where the prior of model A is biased systematically high over a given region, while the prior of model B is biased systematically low. Both models converge to the same response, but when analyzing the increment (yijk(s)- yijk (0)(s)), model A's would be negative and model B's would be positive. If the biases were equally biased, the "consensus flux anomaly" of this ensemble of two would be zero. This is written in the text, e.g. in lines 283-284: "Noting again that the analysis is carried out the inversions' deviations from their respective priors, the map depicts a consensus flux anomaly from the collection of inversions analyzed." How can we interpret a "consensus flux anomaly" when the models are starting from different priors? (This does not apply to the first part of the study, only the MIP results.)**

We appreciate the thorough exposition of this comment. The reviewer has outlined some of the

key challenges in the interpretation of some of the ANOVA components when analyzing the flux increments, particularly for the OCO-2 MIP examples. There are pros and cons to analyzing the flux increments with this approach, and we have further outlined the motivation for this choice in the definition of the functional ANOVA model in Section 3. Statistical modeling traditionally introduces spatio-temporal correlation after accounting for systematic patterns in space and time, and the prior fluxes are a reasonable representation of these systematic structures. For the flux MIP examples, we have noted in the revised manuscript that the inferred model effects are a combination of imperfect transport, prior flux differences, and other algorithm and processing artifacts. We also note that these estimated model effects are useful as a diagnostic for individual modeling groups within the MIP collection. Finally, the estimated overall mean field does exhibit mixed results, but some regions exhibit mean increments in the presence of prior fluxes of the same sign.

- **Going back to the theoretical model A and model B from before: these increments would presumably show up as negative and positive flux model differences with a large spatial range, perhaps similar to the all-red (Ames) and all-blue (CMS-Flux) posterior mean maps seen for $\alpha$ for JJA in North America (Figure 7). However, the authors attribute this to "large-scale regional flux differences among inversion systems with different driving atmospheric transport" (L275-276). This could be the case, but without having common priors, the effects of different priors and different transport cannot be teased apart using this method. Ideally, such an analysis would be carried out on inversion results with identical priors, so that we could truly say something about the influence of the transport model and optimization. Lacking that, this limitation needs to be made completely clear, so that the reader is not misled.**

  This is a further important observation about the use of the flux increments in the analysis. In the revised manuscript, we have added further exposition on this point in the discussion of the methodology, results, and conclusions. Specific revisions include:

  - Functional ANOVA (Section 3): The revised manuscript includes an additional paragraph after Equation 1. This discussion provides additional methodological and practical motivation for analyzing the flux increments. The different MIP modeling groups' processing resulted in small-scale artifacts in prior and posterior fluxes that were not as evident in the flux increments. Further, as noted in a previous comment, large-scale spatial and temporal patterns are common in the prior and posterior fluxes, and common intra-seasonal signals can be uncovered in the flux increments.
  - Results (Section 4): The disussion of the OCO-2 flux MIP results has been modified. In discussing Table 4, we have added a remark, "Further, since the statistical model is applied to the flux increments, prior flux differences contribute to the model effects." In addition, discussion of Figure 6 later, we have added, "the patterns can be a combination of difference in prior fluxes, as well as other aspects of the inversions, particularly atmospheric transport."
  - Conclusions (Section 5): The revised manuscript includes an additional paragraph discussing the limitations of analyzing the flux increments in the MIP experiments. This discussion mentions that the ANOVA model effects could result from combinations of differences in transport, prior, and additional inversion system implementation choices. Some additional remarks on multi-model experiment design, particularly implementing common priors, are included.

  Finally, we have adopted the reviewer's use of the term *flux increment* in discussing this approach in the revised manuscript.

- **As a more minor/technical note: The treatment of the different spatial resolutions in the MIP part of the study should be explicitly explained. For the first part of the study, only results from the CMS-Flux model are used, and the plots appear to match the 4°x5°**

resolution of the model. For the MIP, models of different resolutions are compared (but none with a higher resolution than the still relatively coarse $4° \times 5°$ resolution of CMS-Flux), but the data appear to have been resampled onto a higher spatial grid in Figures 6, 8 and 9. In Figures 7, 10 and 11, the resolution appears smoother still. What procedure was followed?

The revised manuscript includes additional details about the spatial resolution for the flux estimates. These details are provided in the dataset descriptions in Section 2. The fluxes from the fused $CO_2$ experiments with the CMS-Flux system are available at $4° \times 5°$ resolution.

For the OCO-2 flux MIP, the modeling groups all provided gridded fluxes at $1° \times 1°$ resolution, and this is noted at the end of Section 2 in the revised manuscript. These datasets on a common grid are the products that have been provided to the community from the flux MIP and were the data source for the functional ANOVA. Modeling groups implemented their own re-gridding approaches, and this is another potential contributor to model effects in the functional ANOVA. As noted in a previous response and in Section 3 of the revised manuscript, exploratory analysis indicated that the flux increments exhibited fewer local artifacts due to re-gridding.

Finally, the smooth appearance in Figures 7, 10, 11 is mostly the result of the relatively large spatial ranges estimated for the model and data source effects for both MIP functional ANOVA examples. The plots are produced at the $1° \times 1°$ resolution in the same fashion as other maps from the MIP examples.

**2.2 Minor Comments**

- **L1-2: This sentence is a bit odd, making it sound as if measurements made today (present tense) span several years. Perhaps: "has now produced atmospheric greenhouse gas concentration estimates covering a period of several years"?**

  This change has been made in the revised manuscript.

- **L8: Not clear what is meant by "mode differences" here.**

  The term has been removed from the updated abstract to provide additional clarity on the results of the CMS-Flux example over Eurasia.

- **L10: For inversions over different continents, or for fluxes over different continents?**

  Fluxes is the appropriate word. This has been changed in the revised manuscript.

- **L17: The language here makes it sound as if the satellites are providing an estimate of global $CO_2$ concentrations frequently, which is not the case. Instead of "frequent", perhaps quantify it? How many good quality measurements per year, on average?**

  We have modified this discussion slightly in the revised manuscript to address this and the following comment. We have included information on the annual data volume for OCO-2.

- **L18: Has flux inversion become demonstrably more tractable (not quite sure what that means here) and precise due to satellite measurements? If so, please provide a reference providing evidence of this!**

  This was unintentional phrasing in the original manuscript. Our primary intention was to provide some context for contrasts in the pre-processing/use of satellite data versus in situ observations in flux inversions. The revised manuscript provides additional emphasis on this aspect by mentioning the use of spatially-aggregated satellite products in global inversions.

- **L24-25: Not clear what is meant by "dependence can be exploited in quantifying uncertainty". Dependence of what on what? Please explain.**

This was intended to be spatio-temporal dependence. It has been rephrased as "characterizing spatio-temporal correlation is necessary in quantifying uncertainty" in the revised manuscript.

- **L39: collection Earth → "collection of Earth"**

  This change has been made in the revised manuscript.

- **L41: Hyphenate "multiple-component" as a compound adjective, also "spatially-aggregated" on L45.**

  This change has been made in the revised manuscript.

- **L47: with associated → "with an associated"**

  This change has been made in the revised manuscript.

- **L55: Is the year missing from Sain et al. reference here? Is this consistent with the Copernicus style guide?**

  The year for the reference has been added in the revised manuscript.

- **L56: Would replace the comma with a period here, breaking into two sentences. (Or at least a semicolon.)**

  This has been split into two sentences in the revised manuscript.

- **L57: Would be easier to parse if "used in the current work" were preceded and followed by a comma.**

  The commas have been added in the revised manuscript.

- **L70: Which version of ACOS?**

  In the revised manuscript, we have noted that the examples in this paper are based on the OCO-2 Version 9 products.

- **L73-75: Please clarify about the creation of super-observations through aggregation. How it's written here, it sounds as if the data are aggregated across the whole orbit. Aggregation to superobservations is not the same as what is generally considered a L3 product, which is rather gap-filled concentration fields, perhaps through data assimilation to optimize atmospheric state (instead of fluxes). This is explained further in 2.1, but seems inconsistent with what is written here.**

  This description was somewhat unclear as written, and in the revised manuscript, we have clarified that all of the aggregation approaches used produce estimates over some type of regular grid, which is $1° \times 1°$ in the case of the fused products.

- **L101: What is SSDF? I believe this has not been defined.**

  We intended to drop the SSDF acronym from the manuscript. The revised manuscript has removed this item.

- **L103: variables of interest?**

  This change has been made in the revised manuscript.

- **L120: Perhaps specify that you mean ACOS version 9?**

  We have added this note in the revised manuscript.

- **Caption Figure 2: remove "based"**

  This change has been made in the revised manuscript.

- **L181: and: should maybe be changed to ", which" or "that"?**

  We have changed the word choice to "which" in the revised manuscript.

- **L224: parameters → parameters'**

  This change has been made in the revised manuscript.

- **L225: are → is**

  This change has been made in the revised manuscript.

- **L227: is → are**

  This change has been made in the revised manuscript.

- **L231: is → was**

  This change has been made in the revised manuscript.

- **L232: The wording here is a bit dense. Perhaps better: "which makes the MCMC algorithm more computationally efficient."**

  This change has been made in the revised manuscript.

- **Caption Table 3: Not clear what "in the records of [the] fused CO2 experiment" means here. Perhaps just "for the fused CO2 experiment"? Also, "of 95% posterior credible intervals" → "of the 95% credible intervals of the posterior". The same comment applies to the caption of Table 4 as well. In general, I was a bit confused by the use of "the records of fused CO2 experiment", also in the captions of Figures 4 and 5.**

  We have changed the description to "fused $CO_2$ experiment" in the revised manuscript. "Records of fused $CO_2$" is an abbreviation of the overarching $CO_2$ data fusion project, but we agree that the phrase can be confusing out of context. The wording for the credible intervals has been changed in the revised manuscript.

- **Figure 4: Units are missing. Somewhere in the caption it should be made clear that the "mean" is actually the mean difference from the prior for JJA over 2015-2016, rather than the mean flux itself. The label on the lower right panel should be "Interaction" rather than "Interact".**

  The panel label has been changed in the revised manuscript. In addition, the figures now include units for fluxes and flux increments, and the information is included in figure captions.

- **L270: Differences in behaviour, or differences in data coverage?**

  Yes, differences in data coverage conveys the message more clearly. We have made this change in the revised manuscript.

- **L283: Missing the word "on" (carried out on the inversions)?**

  Yes, this has been corrected in the revised manuscript.

- **L315-316: Really? The probability that the difference across years was larger than the aggregation method effect did not exceed 0.6. This does not convince me that using such a fused product provides internally consistent information for analyzing e.g. interannual variability in fluxes. Perhaps I have misunderstood something.**

  This is a fair point, particularly on the interannual effect aspect. In the revised manuscript, we have modified this discussion to highlight the contrast in spatial coherence of these components. We have further noted the challenge of detecting year-to-year differences of the magnitude seen in this

example. Even so, the estimated year-to-year differences can stand on their own as an interannual difference averaged across the aggregation methods. With respect to the aggregation methods, our intent is to characterize the differences due to the method and to highlight impacts on additional inferences from the flux estimates.

- **L339-340: Is the word "variance" (or something else) missing? i.e. "Alternative parameterizations with heterogeneous variance across space could be developed."**

  Yes, this has been corrected in the revised manuscript.

- **Supplement L32: model → models**

  This change has been made in the revised supplement.

**References**

Cressie, N., Bertolacci, M., & Zammit-Mangion, A. (2022). From many to one: Consensus inference in a MIP. *Geophys. Res. Lett.*, *49*, e2022GL098277. doi: 10.1029/2022GL098277

---

## Author Response (AR2)

**Manuscript: Functional ANOVA for Carbon Flux Estimates from Remote Sensing Data**

J. Hobbs, M. Katzfuss, H. Nguyen, V. Yadav, J. Liu

December 2023

**1 Editorial Response**

The topic editor report included some remarks that we address below

- **The manuscript briefly mentions ""functional analysis of variance (ANOVA)" for identifying important flux anomalies, but lacks a detailed explanation of the spatial-temporal statistical model's construction and application. It is recommended that the authors provide a clearer methodological framework for functional ANOVA, especially in the context of its operationalization within spatial-temporal statistical models for estimating common flux signals.**

  We appreciate the reviewer's request for additional clarification on the functional ANOVA methodology. The statistical model's construction is outlined in detail in Section 3 of the manuscript with additional exposition in the supplement. Further, the implementation and application are discussed in subsequent sections. However, we agree that other sections of the manuscript and abstract could set the stage and reiterate the key methodological aspects more effectively. Therefore, the revised manuscript includes an additional sentence in the abstract that elaborates on the method. Additional clarifying statements are included in the Introduction and Conclusions.

- **The distinction between Level 2 and Level 3 products and their specific impact on flux estimation should be more thoroughly explained. It would be beneficial if the authors could conduct additional analyses to elucidate the impact of data product levels on the inversion results, thus strengthening the manuscript's argument.**

  In the revised manuscript, we have clarified the discussion of the product levels in the Introduction. As noted in Section 2.1, none of the inversions studied actually ingest Level 2 products at their native resolution. Thus, the comparisons performed in our study, particularly for the fused $CO_2$ experiment, are comparisons of different approaches to constructing Level 3-type products.

- **While the manuscript acknowledges the influence of multiple sources of uncertainty on the flux solution, it could provide a more explicit description of how these uncertainties are integrated into the spatial-temporal statistical method and how the a posteriori uncertainty is represented. Elaborating on how different sources of uncertainty are managed within the method is recommended.**

  In this work, the partitioning of multiple sources of variability is achieved by defining different factors in the functional ANOVA implementation. Their relative importance is manifested, in part, through the Gaussian process (GP) standard deviations, as discussed in the Results section. We have added a sentence there to this effect. The Introduction notes that the flux a posteriori uncertainty is not always available, and this is the case for OCO-2 MIP products analyzed. We have added this note in Section 2 of the revised manuscript.

- **The manuscript delves into the influence of various factors on global flux estimation, including atmospheric transport modeling and data sources. However, it does not sufficiently discuss how these factors pertain to specific geographic regions and time scales. It is advisable to conduct more detailed regional and temporal analyses of these influencing factors to comprehensively understand their impacts.**

  Indeed, the impact of these sources of variability on the flux estimates can change in space and time. For this reason, we have studied their impacts through application of the functional ANOVA over three distinct continents, as discussed in Sections 3 and 4. While this is not a comprehensive set of regional analyses, it does span a range of geophysical conditions (tropics, mid-latitudes) and in situ data availability. For temporal coverage, the analysis is constrained in two aspects. First, the flux estimates are provided at monthly resolution; second, the OCO-2 record spans the recent decade. As the satellite $CO_2$ record continues, trends and interannual variability can be studied more thoroughly. In response to this comment, we have included the mutli-continent analysis in the Introduction and added some suggested future work in the Conclusions.

**2  Referee Report 3**

Additional comments from Referee Report 3 are addressed here

- **Modeling the "deviation" $(y - y^{(0)})$ is well motivated, but it is unclear from the paper what $y^{(0)}$ (prior flux) is specifically and how it's estimated/evaluated. Specifically writing this out in Sections 4.1 and 4.2 (or even in the supplement) for the data examined would be helpful.**

  The construction of prior fluxes for inversion systems is an art in its own right. Particularly for the OCO-2 MIP, each inversion system has its own prior flux fields, but these come from a relatively small collection of input sources. The prior fluxes attempt to capture climatological behavior of the biosphere, in combination with fossil fuel emissions and other known perturbations, particularly fires. We have a brief discussion on this in Section 3 of the revised manuscript, and the flux prior maps are available in the supplement.

- **Because the error (epsilon) values are so large, I'd like to see a plot of these. Perhaps they could be made available in the supplement.**

  This is an interesting suggestion, and the functional ANOVA residuals can be assembled from the MCMC results. We include here an example (Figure EC.1) from the OCO-2 MIP analysis for North America. At this point, we do not think this additional exposition is warranted to include these results in the supplement.

- **I'd like more interpretation of the results and their implications.**

  - **An example of a place where more CO-2 and OCO-2 context would be useful is with Figure 10. Page 17 discusses that the estimated model effects for the mean flux increment changes sign across the equator or a "north-south contrast." What are the implications of this for OCO-2? Why does this matter for the MIP? Is this a surprising or expected result? In other words, how should scientists use this new knowledge/understanding?**

    This is an important point, and we have added further discussion on these differences in the revised manuscript, mentioning that north-south transport among the inversion systems could play a role. We also note that the density of OCO-2 observations changes from north to south across Africa.

[Figure]

[Figure]

Figure EC.1: Functional ANOVA estimated residuals for the MIP example over North America. Units are gC m$^{-2}$ yr$^{-1}$

– **An example of cases where more statistical context would be useful is in regards to Figure 11. Figure 11 provides credible intervals and shows that most of the intervals include 0. As noted on page 18, places where the credible inferences do not include 0 are places where there are few/no in situ observation sites. Does it make sense to even make inferences here? How much are inferences at these locations biasing the results at other locations?**

We appreciate this remark as well. In this instance, inversion systems' representation of atmospheric transport could be a factor yet again. The sparse in situ observations combine with the assumed atmospheric transport to yield the flux estimates. In fact, the reviewer's remark is an alternative interpretation of the data source effect; areas with sparse in situ coverage can have a relative "bias" in flux space versus the satellite inversions. We have added further interpretation in the revised manuscript.

• **Finally, the code is a bit overwhelming to try to reproduce the results. The "readme" files are helpful but I recommend having a "main" file where each relevant file is sourced in order to reproduce main results (even if it comes with a computation time warning).**

We agree that there is a learning curve for the code repository. However, a driving script is not practical given the longer runtime for the MCMC sampler steps. With that said, we will update the examples' README to indicate which steps/scripts can be skipped over if a user simply wants to make use of the final MCMC samples that are available on Zenodo.